# Boundary-Consistent Graph Neural Networks for Topological Flux Prediction

## Abstract

Graph Neural Networks (GNNs) have achieved notable success in spatiotemporal modeling across diverse application domains. However, their efficacy in flux prediction (FP), where the goal is to model spatiotemporal fluid transport over networked physical systems, remains contentious. Recent studies report that GNNs can underperform even simple baselines in FP settings, leading to a claim that GNNs may be intrinsically ill-suited for such tasks.

In paper, we revisit this claim by dissecting the GNN learning dynamics on fluid transport networks, with an emphasis on its boundary regions. Specifically, we decompose the graph into boundary and interior nodes, where boundary nodes regulate the total influx and are the primary interface with boundary forcing. Our empirical and theoretical analyses reveal that dominant prediction errors concentrate at boundary nodes. From a dynamical-systems perspective, we interpret the boundary errors as the consequence of missing upstream boundary context, which causes degraded performance on boundaries. We therefore hypothesize that the observed performance degradation of GNNs was not caused by their expressivity; rather, it arises from the deficit of explicit boundary-context closure during training.

To validate this hypothesis, we propose `gTFP`, which learns ghost-node proxies as boundary-context proxies. Each boundary node is augmented with an associated ghost node that serves as a learned boundary-closure variable. This yields a ghost–boundary–interior coupled system, which we solve using an implicit fixed-point formulation. The resulting equilibrium *jointly* infers the boundary-closure variables and propagates it into the interior. This enriches standard GNN backbones with boundary-consistent representations while preserving interior message passing. Extensive experiments on two real-world fluid network datasets demonstrate that `gTFP` improves standard GNNs by reducing average MSE by 8.4% and 5.0%, and boundary-node MSE by 11.2% and 7.1%, respectively. For computational efficiency, we further introduce an explicit inverse-operator solver that amortizes the fixed-point inference and accelerates inference by up to 2×, depending on the backbone architecture.

## 1 Introduction

Flux prediction enjoys broad use in fluid systems (Kratzert et al., 2021; Jin et al., 2023), *e.g.,* flood forecasting (Jiang et al., 2025; Bentivoglio et al., 2025), hydrochemical modeling (Mangold & Tsang, 1991), estuarine circulation (Geyer & MacCready, 2014), among others. Since such systems unfold in both space and time, Graph Neural Network (GNN) (Zhou et al., 2020; Wu et al., 2020) emerges as a seemingly plausible modeling choice, given its demonstrated success in related tasks such as traffic forecasting (Jin et al., 2023) and energy transmission (Varbella et al., 2024).

Yet, recent studies reveal that GNNs often ignore the underlying fluid dynamics and learn absurd patterns from data, *e.g.,* predicting fluxes moving from downstream to upstream, which violate gravity and conservation laws (Kirschstein & Sun, 2024). As a result, some conclude that incorporating fluid system topology offers little benefit, as GNNs underperform even simple baselines like multilayer perceptrons (MLPs), which do not model graph structure at all (Kirschstein & Sun, 2024).

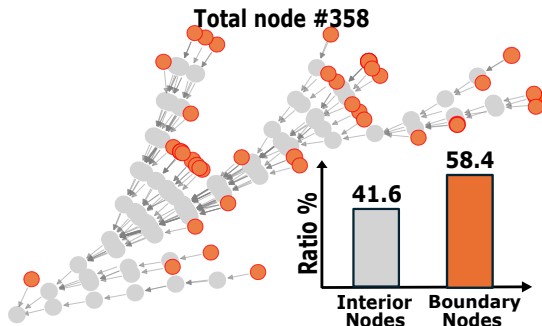

**Avg. MSE per node**

| Model | Boundary | Interior | Diff. (%) |
|---|---|---|---|
| ResGCN | 0.1408 | 0.0985 | 30.04% |
| ResGAT | 0.1401 | 0.0836 | 40.33% |
| GCNII | 0.1511 | 0.0891 | 41.03% |
| GNN error (avg.) | 0.1217 | | – |
| MLP | 0.1210 | 0.1030 | 14.89% |
| MLP error (avg.) | 0.1135 | | – |

Figure 1: A fluid network from the LamaH-CE2 dataset, where boundary nodes account for more than half of the graph topology. Although standard GNNs outperform a simple baseline (*e.g.,* MLP) on interior nodes, a misleading conclusion may arise due to their larger loss on boundary nodes, resulting in a higher overall average MSE. We hypothesize that this is because GNNs lack an explicit boundary-context closure mechanism, which is essential in classic PDE-based flux prediction models for networked fluid systems.

In this paper, we argue that such conclusions are premature. We revisit the prediction loss patterns from GNN-based flux models and find that the dominant errors lie at the *boundary nodes* that interface with unobserved upstream boundary context. As Figure 1 illustrates, whereas boundary nodes account for more than half of the total nodes in a fluid network, existing models consistently yield much higher prediction errors on them compared to interior nodes. In fact, if we restrict evaluation to interior nodes alone, GNN predictors significantly outperform baseline MLPs. This discrepancy suggests that treating all nodes equally without distinguishing boundary from interior may underlie the perceived failure of GNNs in fluid systems.

To scrutinize this observation further, we analyze the GNN learning behavior from a dynamical-system perspective (Poli et al., 2019), where message-passing simulates the update of node states under local interaction rules (*e.g.,* a differential equation), while each such layer proceeds a discrete time step (*e.g.,* an Euler step). Boundary conditions in such systems constrain their solution space (LeVeque, 2007)(in our setting, this corresponds to exogenous inflow forcing at upstream sources). In GNN-based flux models, boundary nodes are discrete counterparts of these conditions, as they regulate influx into the entire network through boundary forcing. We hypothesize that missing upstream boundary context creates a boundary-side information deficit that can degrade GNN predictions. Our goal is not to enforce global PDE constraints, but to learn boundary-closure variables and reduce prediction errors at boundary nodes.

To instantiate and validate our hypothesis, we propose a novel computing framework, termed **gTFP**, where we borrow the ghost node technique (Tseng & Ferziger, 2003) from finite-difference methods (FDM) to implement learned boundary-context proxies. Specifically, we approximate ghost-node embeddings from boundary nodes and their immediate (downstream) neighbors. This imposes recursive coupling on message-passing, where each boundary node depends on ghost nodes, whose embeddings are in turn defined by interior and boundary nodes themselves. Unknown node representations thus appear on both sides of the update equations, making standard layer-wise GNN training inapplicable. To solve this, we adopt implicit GNN (Gu et al., 2020), which recasts message-passing as a fixed-point problem and seeks an equilibrium that aligns node representations with the structural relationships determined by boundary conditions.

Note, imposing different boundary conditions on fluid system may derive disparate structural couplings among ghost, boundary, and interior nodes. Each such coupling defines a unique augmentation of graph adjacency, which entails extensive craftsmanship to adapt the implicit GNN solver to every possible augmentation. To counter this, we further unify our framework by treating the inverse of the augmented adjacency as a learnable operator, lending a closed-form approximation to the implicit solution.

**Specific contributions** in this paper are summarized as follows.

(1) We decouple error sources in GNN-based flux models and find the dominant loss stems from missing explicit boundary-context closure. The analysis is presented in Section 3.

(2) We propose `gTFP` to learn boundary-context proxies implemented as ghost-node embeddings; it outperforms GNN baselines by 6.68% on average and mitigates the boundary–interior loss gap by 11.03% on two real datasets. The details of `gTFP` are documented in Section 4.1, with its corresponding analysis in RQ.1 and RQ.2 of Section 5.

(3) We devise a unified operator-learning view within `gTFP` that avoids hand-crafted adjacency in implicit solvers, yielding up to $2\times$ training speedup (backbone-dependent) without sacrificing accuracy. The design of this solver is in Section 4.3, with its comparative performance in RQ.3 – RQ.5 of Section 5.

## 2 Preliminaries

**Notation and Problem Statement.** Let $G = (V, E)$ denote a directed fluid network, where nodes $V$ represent local observation points and edges $E$ indicate the direction of flow. We define the adjacency $\mathbf{A} \in \{0,1\}^{|V| \times |V|}$, such that $\mathbf{A}_{i,j} = 1$ if there exists a directed edge from node $v_i$ to $v_j$, and $\mathbf{A}_{i,j} = 0$ otherwise. Note that $\mathbf{A} \neq \mathbf{A}^\top$. A node $v_j$ is said to be a neighbor of $v_i$, denoted $v_j \in \mathcal{N}(i)$, if $\mathbf{A}_{j,i} = 1$.

At time $t$, each node $v_i \in V$ is associated with a feature matrix $\mathbf{h}_i \in \mathbb{R}^{W \times d}$, which stores $d$ physical measurements (*e.g.,* velocity, pressure, slope) over a historical window of $W$ time steps. Stacking features across all nodes yields a tensor $\mathbf{H} = [\mathbf{h}_1, \dots \mathbf{h}_{|V|}]^\top \in \mathbb{R}^{|V| \times W \times d}$. The goal of flux prediction is to learn a predictive model $f$ that forecasts a target physical quantity (*e.g.,* flux volume) at a future step $t + n$, with $n$ the prediction horizon. Let $\mathbf{y} \in \mathbb{R}^{|V|}$ denote the ground-truth values of this target quantity. Our learning objective is to minimize the empirical loss $\ell(\mathbf{y}, f(\mathbf{H}, \mathbf{A}))$.

We define the *boundary nodes* as a general set $V_{\mathrm{BN}} \subseteq V$ specified by the graph directionality, geometric location, or the underlying physical system. Intuitively, boundary nodes are the primary interface through which boundary forcing is prescribed or enters the system. In the directed fluid networks studied in our main experiments, we instantiate this boundary set as the zero in-degree nodes, *i.e.,* $V_{\mathrm{BN}} = \{v_b \in V \mid \deg^-(v_b) = 0\}$. In this directed setting, these nodes lie at the most upstream points of $G$ and regulate the influx into the whole system. The remaining nodes are referred to as *interior nodes*, defined as $V_{\mathrm{IN}} = V \setminus V_{\mathrm{BN}}$.

## 3 Problem Analysis

A paradoxical observation is that if the predictive model $f$ is instantiated as GNN, it often underperforms simple baselines (*e.g.,* MLP) and shows little difference between predictions computed from $f(\mathbf{H}, \mathbf{A})$ and $f(\mathbf{H}, \mathbf{A}^\top)$. Here, as $\mathbf{A}$ encodes the ground-truth forward fluid flow, $\mathbf{A}^\top$ represents a physically-implausible, reversed flow. Kirschstein & Sun (2024) concludes that GNNs may not work in flux prediction tasks, as they fail to distinguish directional flow consistency, which is fundamental in physical systems.

### 3.1 GNNs as Neural Differential Equations.

We argue that such a conclusion is premature and advocate an alternative interpretation through the lens of numerical solvers (Liu et al., 2025). Write a standard message-passing update as

$$\mathbf{h}_i^{(l+1)} = \mathbf{h}_i^{(l)} + \sum_{v_j \in \mathcal{N}(i)} \psi^{(l)}(\mathbf{h}_i^{(l)}, \mathbf{h}_j^{(l)}), \quad \mathbf{h}_i^{(0)} = \mathbf{h}_i, \tag{1}$$

where $\mathbf{h}_i^{(l)}$ denotes the representation of node $v_i$ at layer $l$, and $\psi^{(l)}$ is the message function aggregating information from its upstream neighbors $v_j \in \mathcal{N}(i)$. We view Eq. (1) as an explicit integration scheme, that mimics a partial differential equations (PDEs) solver (LeVeque, 2007), such as

$$u^{t+1}(x_i) = u^t(x_i) + \Delta t \cdot F(u^t(x_i), u^t(x_{i+1})), \tag{2}$$

$$\text{s.t. } u^t(x_{i+1}) \approx B(u^t(x_i), \partial_x u^t(x_i)), \quad \forall v_i \in \mathcal{V}_{BN}. \tag{3}$$

where $u^t(x_i)$ is the state of a physical quantity (*e.g.,* flux volume) at location $x_i$ and time $t$, and $F$ represents the spatial derivative or transport dynamics. Consider, for example, a discretized advection equation (Chock, 1991) that implements $F(u^t(x_i), u^t(x_{i+1})) = \frac{\partial}{\partial x}(u^t(x_{i+1}) - u^t(x_i))$, which models the propagation of the

quantity through its upstream neighbor $x_{i+1}$. We can observe an algebraic similarity between Eq. (1) and Eq. (2), where the GNN layer index $l$ parallels the time step $t$, and the message function $\psi$ acts as the discrete derivative $F$ across the graph topology. At boundary locations, Eq. (3) uses a boundary operator $B(\cdot)$ to approximate the out-of-domain upstream value required to evaluate $F$.

A natural question: if PDE solvers operate robustly and in a topology-aware manner for fluid systems, and message-passing mimics their computational structure, then *why do GNN empirically fail on the same task?*

**Boundary-Context Closure Deficit.** We hypothesize that message passing lacks an explicit mechanism to close the missing boundary context at upstream boundary nodes; we refer to this as a boundary-context closure deficit. PDE solvers explicitly enforce such conditions, as seen in Eq. (3), which closes the missing upstream state so that the flux term $F$ in Eq. (2) is well-defined at the boundary location $x_b$. Intuitively, when $x_b$ lies at the spatial boundary, its upstream neighbor $x_{b+1}$ is undefined, invalidating the computation of the flux term $F$ in Eq. (2) that depends on $u^t(x_{b+1})$. The boundary condition in Eq. (3) complements this by approximating the out-of-domain upstream value from the local state $u^t(x_b)$. To wit, we instantiate the boundary operator $B(\cdot)$ in Eq. (3) with a Robin-type boundary condition (Busse et al., 2017):

$$u^t(x_{b+1}) = \omega_1 \cdot u^t(x_b) + \omega_2 \cdot \partial u^t(x_b)/\partial x, \tag{4}$$

with $\omega_1, \omega_2 \in \mathbb{R}$, which closes the dynamical system by postulating an interpolation between $x_b$ and its spatial derivative to approximate the undefined, out-of-boundary $x_{b+1}$.

In contrast, GNNs lack an explicit closure mechanism for missing upstream boundary context at graph boundaries. For a boundary node $v_b$ which, by definition, has no incoming edge and thus no upstream neighbor, namely $\mathcal{N}(b) = \emptyset$. As a result, the boundary node has no upstream inputs to reflect upstream boundary context, Eq. (1) collapses to $\mathbf{h}_b^{(l+1)} = \mathbf{h}_b^{(l)} + \psi^{(l)}(\mathbf{h}_b^{(l)}, \mathbf{0})$, meaning that the update of $v_b$ depends solely on its own features and receives no information from the graph topology. This isolation over successive layers leads to degraded boundary node embeddings and, eventually, to substantial prediction errors.

## 3.2 Empirical Validation.

To validate our hypothesis, we analyze and compare the prediction losses of ResGAT (Residual Graph Attention Networks) and MLP by separating the errors incurred at boundary versus interior nodes. The results are summarized in Table 1.

We make two observations from these results. ***First***, the overall mean squared error (MSE) misleadingly suggests that the GNN underperforms an MLP (.1166 > .1135), when in fact the GNN performs substantially better in regions where it can leverage graph topology. The failure of GNN is mainly attributed to the boundary nodes, incurring a .1401 MSE. This large boundary loss obscures the otherwise strong performance of GNN-based flux prediction on interior nodes, where MSE drops to .0836. ***Second***, the seemingly similar overall losses

Table 1: Node-wise MSE Comparison

| Node Type | MSE For. (A) | MSE Rev. (A$^\top$) |
|---|---|---|
| Boundary ($V_{\text{BN}}$) | .1403±.0010 | .1352±.0011 |
| Interior ($V_{\text{IN}}$) | .0835±.0007 | .0940±.0008 |
| All nodes ($V$) | .1168±.0008 | .1177±.0009 |
| MLP | | .1136±.0007 |

using the ground-truth forward flow $\mathbf{A}$ (.1166) and the reverse flow $\mathbf{A}^\top$ (.1179) are deceptive. The forward model is disproportionately penalized by boundary node errors, which suppress its average performance and mask its superiority over the reversed model. When focusing on interior nodes only, the forward model achieves an MSE of .0836, outperforming the reversed model at .0939. This aligns with physical intuition and demonstrates the positive impact of graph topology on learning meaningful representations in regions where directional flow information is available.

Note, these MSE results are normalized, where .001 change means 1% flux volume change. For a mid-size river network (Discharge: $100\text{m}^3/\text{s}$), a .01 MSE error in discharge prediction equals a daily volume discrepancy of $86,400\text{m}^3$, which equals to $\sim$35 Olympic pools. This may cause critical failure with cascading consequences, *e.g.,* threaten the survival of aquatic species (Poff et al., 1997), lead to dangerous underestimations of pollution risk (Whitehead et al., 2009), and be amplified into financial losses for hydropower and navigation (Lehner et al., 2005; Jonkeren et al., 2007).

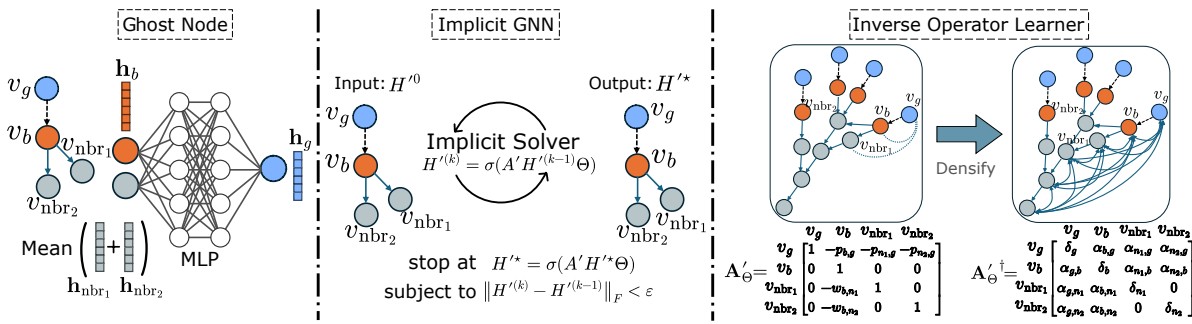

Figure 2: Overview of the proposed `gTFP` framework. Left: Ghost-node construction (Section 4.1). For each boundary node $v_b$, we introduce a corresponding ghost node $v_g$. Its embedding $\mathbf{h}_g$ is learned from the boundary embedding $\mathbf{h}_b$ and the aggregated embeddings of its downstream interior neighbors $\mathbf{h}_{\mathrm{nbr}}$. Middle: Implicit solver (Section 4.2). We perform implicit message passing on the augmented graph with ghost nodes using a fixed-point solver, converging to $\mathbf{H}'^\star$. Shared parameters $\Theta$ are applied consistently across updates. Right: Explicit inverse-operator learning (Section 4.3). We learn an explicit inverse operator on the densified adjacency $\mathbf{A}'_\Theta{}^\dagger$, enabling layer-wise updates through a learnable inverse mapping.

## 4  The `gTFP` Approach

This section presents our `gTFP` framework. Standard message-passing GNNs do not explicitly model upstream boundary influence, often yielding degraded accuracy near system boundaries. To address this, we introduce ghost nodes as learned boundary-context proxies. We then solve the induced coupled boundary–interior update using either an implicit or an explicit solver. In Section 4.1, we describe how to construct and learn ghost-node embeddings from surrounding interior nodes to serve as learned boundary-closure variables. Section 4.2 formulates an implicit solver that integrates ghost nodes into a fixed-point framework inspired by numerical PDE methods. Section 4.3 further provides a computationally efficient, explicit counterpart by learning a dense inverse operator, allowing layer-wise updates that preserve boundary-awareness while avoiding the cost of fixed-point iteration.

### 4.1  Learn Boundary-Context Proxies via Ghost Nodes

To remedy the boundary-context closure deficit in GNNs, we learn boundary-context proxies from data via a numerical ghost-node construction (Tseng & Ferziger, 2003). Let $v_g$ denote a virtual ghost node corresponding to a boundary node $v_b$, and $v_{\mathrm{nbr}}$ denote the interior (downstream) neighbor of $v_b$, such that $v_{\mathrm{nbr}} \in \{v_j \mid v_b \in \mathcal{N}(j)\}$. We draw insights from PDE solvers to learn the embedding of $v_g$ from the coupling among $v_g$, $v_b$, and $v_{\mathrm{nbr}}$. Specifically, we instantiate the boundary operator $B(\cdot)$ in Eq. (3) with a Robin-type boundary condition (Busse et al., 2017) and discretize it to derive (we compare Dirichlet & Neumann variants in App. A.1):

$$\mathbf{h}_g^{(l)} = \omega_1 \cdot \mathbf{h}_b^{(l)} + \omega_2 \cdot (\mathbf{h}_b^{(l)} - \mathbf{h}_{\mathrm{nbr}}^{(l)})/\Delta x, \tag{5}$$

where $\mathbf{h}_{\mathrm{nbr}}^{(l)}$ is the embedding of $v_{\mathrm{nbr}}$ at the $l$-th layer. In implementation, we parameterize $\omega_1$ and $\omega_2$ via an MLP, defined as $\mathbf{h}_g^{(l)} = \mathrm{MLP}(\mathrm{Concat}(\mathbf{h}_b^{(l)}, \mathbf{h}_{\mathrm{nbr}}^{(l)}); \theta_{\mathrm{gh}})$. Main steps for ghost-node learning are presented in Algorithm A.10.1 (in App. A.10).

The total number of such ghost nodes equates to boundary nodes. Defining the set of ghost nodes $V_{\mathrm{GH}} = \{v_g\}$, we have $|V_{\mathrm{GH}}| = |V_{\mathrm{BN}}|$. To proceed message-passing, we define a graph augmentation operator $\mathcal{A}$, which takes the original graph as inputs and augments it with ghost nodes as follows.

$$(\mathbf{A}', \mathbf{H}') = \mathcal{A}(\mathbf{A}, \mathbf{H}), \tag{6}$$

where $\mathbf{H}' = [\mathbf{H}, \{\mathbf{h}_g\}_{|V_{\mathrm{BN}}|}]^\top \in \mathbb{R}^{(|V|+|V_{\mathrm{GH}}|) \times W \times d}$ denotes the augmented node feature matrix, and $\mathbf{A}' \in \{0,1\}^{(|V|+|V_{\mathrm{GH}}|) \times (|V|+|V_{\mathrm{GH}}|)}$ is the augmented adjacency, such that $\mathbf{A}'_{i,j} = 1$ if $v_i$ is an (upstream) neighbor of $v_j$, namely $v_i \in \mathcal{N}(j)$, and $\mathbf{A}'_{i,j} = 0$ otherwise.

The augmentation in Eq. (6) connects each ghost node to a downstream boundary neighbor in a deterministic way, which challenges layer-by-layer message passing. Specifically in numerical methods, to improve

numerical stability one often adopts implicit time discretization, which enforces the Robin-type boundary condition via ghost nodes through a time-coupled relation involving both boundary and ghost states at the new time level (Tseng & Ferziger, 2003), e.g.,

$$(1 - \frac{\delta_1^2 \Delta t}{\delta_2})u^{t+1}(x_b) + (\frac{\delta_1 \Delta t}{\delta_2})u^{t+1}(x_g) = u^t(x_b), \tag{7}$$

where $\delta_1, \delta_2 \in \mathbb{R}$ are two physical coefficients. The derivation from Eq. (2) and Eq. (3) to Eq. (7) are deferred to App. A.2 due to the space limit. Drawing analogy to Eq. (7), learning the ghost node embeddings is constrained by $\alpha_1 \cdot \mathbf{h}_b^{(l+1)} + \alpha_2 \cdot \mathbf{h}_g^{(l+1)} = \mathbf{h}_b^{(l)}$, $\exists \alpha_1, \alpha_2 \in \mathbb{R}$, resulting in the message-passing on boundary node as:

$$\mathbf{h}_b^{(l+1)} = \mathbf{h}_b^{(l)} + \sum_{v_g \in \mathcal{N}(b)} \psi^{(l)}(\mathbf{h}_b^{(l)}, \mathbf{h}_g^{(l+1)}) \tag{8}$$

As the implicit update in Eq. (8) shows, computing the boundary embedding $\mathbf{h}_b^{(l+1)}$ requires the ghost embedding $\mathbf{h}_g^{(l+1)}$. Meanwhile, we enforce the ghost-proxy relation Eq. (5) at layer $l+1$ to close the boundary condition within the same layer update, which makes $\mathbf{h}_g^{(l+1)}$ depend on $\mathbf{h}_b^{(l+1)}$ in return. Together, $\mathbf{h}_b^{(l+1)}$ and $\mathbf{h}_g^{(l+1)}$ are jointly determined by a coupled system, which we reformulate in the next subsection to motivate an implicit fixed-point solver.

To proceed, we can rewrite the analogous implicit update for interior nodes from Eq. (8). For interior nodes $\forall v_i \in V_{\text{IN}}$, they are governed by the standard implicit scheme as

$$\alpha_1 \mathbf{h}_i^{(l+1)} - \alpha_2 \mathbf{h}_{i+1}^{(l+1)} = \mathbf{h}_i^{(l)}, \tag{9}$$

where $\mathbf{h}_i^{(l+1)}$ and $\mathbf{h}_{i+1}^{(l+1)}$ are the embeddings of node $v_i$ and its upstream neighbor $v_{i+1}$ at the $(l+1)$-th layer, respectively. Moreover, rearranging the ghost proxy in Eq. (5) into an implicit form yields

$$\left(\omega_1 + \frac{\omega_2}{\Delta x}\right)\mathbf{h}_b^{(l+1)} - \left(\frac{\omega_2}{\Delta x}\right)\mathbf{h}_{\text{nbr}}^{(l+1)} - \mathbf{h}_g^{(l+1)} = 0. \tag{10}$$

Assembling these equations yields the unified augmented system:

$$\begin{pmatrix} \alpha_1 & 0 & \cdots & 0 & \alpha_2 \\ -\alpha_2 & \alpha_1 & \cdots & 0 & 0 \\ \vdots & \ddots & \ddots & \vdots & \vdots \\ 0 & \cdots & -\alpha_2 & \alpha_1 & 0 \\ \omega_1 + \frac{\omega_2}{\Delta x} & \frac{\omega_2}{\Delta x} & \cdots & 0 & -1 \end{pmatrix} \cdot \begin{pmatrix} \mathbf{h}_b^{(l+1)} \\ \mathbf{h}_{M[b]}^{(l+1)} \\ \vdots \\ \mathbf{h}_1^{(l+1)} \\ \mathbf{h}_g^{(l+1)} \end{pmatrix} = \begin{pmatrix} \mathbf{h}_b^{(l)} \\ \mathbf{h}_{M[b]}^{(l)} \\ \vdots \\ \mathbf{h}_1^{(l)} \\ 0 \end{pmatrix}, \tag{11}$$

where $M[b] \in \mathbb{N}$ denotes the number of interior nodes on the branch which starts at a boundary node $v_b$. The indices $M[b], \ldots, 1$ enumerate the interior nodes along this branch, starting from the interior node $v_{M[b]}$, being an immediate neighbor of $v_b$, and following the flow direction toward the farthest end node $v_1$. For multiple upstream branches, we assemble one such block in App. A.3. This reduces Eq. (11) to the compact form $\mathbf{A}' \mathbf{H}'^{(l+1)} = \mathbf{H}'^{(l)}$, with $\mathbf{A}' \in \mathbb{R}^{(M[b]+2) \times (M[b]+2)}$. This leads to a coupled system (LeVeque, 2007), necessitating an implicit solver via fixed-point iteration.

## 4.2 Implicit Fixed-Point Solver

We treat the ghost-augmented boundary–interior coupling in Eq. (11) as a coupled system and solve it using an implicit fixed-point solver (i.e., Implicit GNN (Gu et al., 2020)). This step jointly updates the boundary and interior embeddings by solving for a consistent equilibrium, similar to implicit schemes in numerical PDE solvers, and serves as a benchmark for the explicit inverse-operator solver in Sec. 4.3. We define an equilibrium mapping on the ghost-augmented embeddings $\mathbf{H}'$ and use its fixed point as the implicit update.

At $l+1$, we learn the predictor and solver parameters by minimizing the prediction loss on the *original* nodes:

$$\min_{f,\Theta,\theta_{\mathrm{gh}},C} \ell\Big(\mathbf{y},\, f\Big(\mathbf{H}^{(l+1)}\Big)\Big), \qquad \mathbf{H}^{(l+1)} := \Pi_A\big(\mathbf{H}_\star'^{(l+1)}\big). \tag{12}$$

Here $\Pi_A(\cdot)$ projects embeddings from the augmented space (including ghost nodes) back to the original node indices in $G$. This projection is used because ghost nodes are unlabeled, so supervision is defined only on the original nodes. We use $\mathbf{H}'$ to denote ghost-augmented node features, where the ghost components are produced by the proxy module in Eq. (5) parameterized by $\theta_{\mathrm{gh}}$. Here $\Theta$ and $C$ are learnable parameters in the IGNN equilibrium equation, whose roles are specified in Eq. (13).

To approximately realize the coupled update implied by Eq. (11) within a learnable GNN architecture, we define $\mathbf{H}_\star'^{(l+1)}$ as the fixed point of

$$\mathbf{H}_\star'^{(l+1)} = \sigma\Big(\mathbf{A}'\,\mathbf{H}_\star'^{(l+1)}\,\Theta + C\,\mathbf{H}'^{(l)}\Big), \tag{13}$$

where $\sigma(\cdot)$ is a non-linear activation. The weights $\Theta$ control the embedding transformation in the equilibrium equation, while $C$ conditions the equation on the layer input $\mathbf{H}'^{(l)}$. Eq. (13) provides a learnable fixed-point solver for the same-layer closure of the ghost-augmented boundary–interior coupling, thereby realizing the coupled update implied by Eq. (11).

Since Eq. (13) typically has no closed-form solution, we approximate $\mathbf{H}_\star'^{(l+1)}$ via $K$ inner fixed-point iterations. We initialize $\mathbf{H}'^{(k=0)} = \mathbf{H}'^{(l)}$ and iterate for $k = 0, 1, \ldots, K-1$: $\mathbf{H}'^{(k+1)} = \sigma\big(\mathbf{A}'\,\mathbf{H}'^{(k)}\,\Theta + C\,\mathbf{H}'^{(l)}\big)$. After $K$ steps, we take the numerical equilibrium solution as $\mathbf{H}_\star'^{(l+1)} := \mathbf{H}'^{(K)}$. Here $k$ indexes the inner fixed-point iterations used to compute the equilibrium at $l+1$. The inner loop can stop early when $\|\mathbf{H}'^{(k)} - \mathbf{H}'^{(k-1)}\|_F < \varepsilon$, or stop at the maximum iteration budget $K$. Gradients w.r.t. $\Theta$ and $C$ are obtained through the fixed-point computation, while gradients w.r.t. $\theta_{\mathrm{gh}}$ backpropagate through the ghost components of $\mathbf{H}'$ generated by Eq. (5) (Gu et al., 2020; Chen et al., 2023).

To ensure existence and uniqueness of the fixed point (and hence stable convergence of the inner iterations), we follow (Gu et al., 2020) and impose a Perron–Frobenius (PF) spectral constraint so that the equilibrium mapping is a contraction. Intuitively, if the effective spectral strength induced by $\mathbf{A}'$ and $\Theta$ is too large, the fixed-point iteration may diverge.

To analyze the spectral strength of the bilinear term $\mathbf{A}'\mathbf{H}'\Theta$, we vectorize it as $\mathrm{vec}(\mathbf{A}'\mathbf{H}'\Theta) = \big(\Theta^\top \otimes \mathbf{A}'\big)\mathrm{vec}(\mathbf{H}')$, where $\otimes$ denotes the Kronecker product, which isolates the roles of the augmented $\mathbf{A}'$ and the propagation weights $\Theta$ (Schacke, 2004). Note that the constant input term $C\,\mathbf{H}'^{(l)}$ does not affect contraction since it does not depend on $\mathbf{H}'$. Let $\lambda_{\mathrm{pf}}(\mathbf{A}')$ be the PF eigenvalue (largest eigenvalue) of $\mathbf{A}'$ (Berman & Plemmons, 1994) and let $\|\Theta\|_\infty$ be the maximum absolute row-sum norm, which provides a convex upper bound on its spectral norm (Zheng & Wang, 2008). Using the standard IGNN bound (Gu et al., 2020), we enforce the strict contraction condition $\lambda_{\mathrm{pf}}(\mathbf{A}')\,\|\Theta\|_\infty < 1$, which guarantees a unique fixed point and stabilizes implicit updates. In practice, we adopt the PF solver/normalization strategy in Gu et al. (2020) to maintain this constraint during training. Algorithm A.10.2 (App. A.10) summarizes the main steps.

## 4.3 Explicit Inverse-Operator Solver

While the ghost nodes compensate for the boundary-context closure deficit, and Eq. (13) provides an implicit yet effective solution for them, this solution suffers from two key efficiency limitations. First, as shown in Figure 5a, although the ghost node-enhanced implicit GNN reduces the prediction loss on boundary nodes by 7.2 %, it suffers from high computational cost, running approximately 13× slower than standard layer-wise message-passing GNNs. Second, the structure of the augmented adjacency $\mathbf{A}'$ can vary *w.r.t.* the modeling choice of boundary condition. In practice, different numerical schemes, *e.g.,* Robin-type as we used in Eq. (4), can be used to impose boundary constraint on the same physical system. As such, our implicit solver that assumes a fixed $\mathbf{A}'$ loses flexibility in adapting to such variations to learn ghost node proxies.

These limitations motivate us to tailor a layer-by-layer counterpart to the implicit solver in Eq. (13). By learning ghost nodes, it incorporates boundary-context proxies to mitigate boundary-side errors, while preserving the computational efficiency of a standard GNN.

Observing Eqs. (5), (6), and (8), we note that although $\mathbf{A}'$ connects each ghost node $v_g$ only to its downstream boundary node $v_b$, the computation of $\mathbf{h}_g$ in fact depends on both $\mathbf{h}_b$ and the interior neighbor embeddings $\mathbf{h}_{\mathrm{nbr}}$. This reveals an implicit computational graph that spans $\mathbf{h}_g$, $\mathbf{h}_b$, and $\mathbf{h}_{\mathrm{nbr}}$, lending to consider whether we can construct a weighted adjacency matrix $\mathbf{A}'_\Theta \in \mathbb{R}^{(|V|+|V_{\mathrm{GH}}|) \times (|V|+|V_{\mathrm{GH}}|)}$ that captures these interactions in a layer-wise message-passing regime.

To this end, we decompose the augmented adjacency $\mathbf{A}'$ into two subgraphs. The ***first*** is a standard GNN over the augmented node set $V \cup V_{\mathrm{GH}}$, where each node (whether interior, boundary, or ghost) participates in message passing. We let $w_{i,j}$ denote the message-passing weight from node $v_j$ to $v_i$, with $v_i, v_j \in V \cup V_{\mathrm{GH}}$. The ***second*** is a bipartite subgraph linking ghost nodes $V_{\mathrm{GH}}$ to original nodes $V$, which reflects how ghost nodes are constructed from the boundary condition. Let $p_{i,j}$ denote the interpolation weight from an original node $v_i \in V$ to a ghost node $v_j \in V_{\mathrm{GH}}$, of which the edges are restricted to connect nodes across the bipartition. This allows us to encode various boundary condition types in a unified message-passing framework. For example, write $\mathbf{h}_g$ and $\mathbf{h}_b$ the ghost and boundary node embeddings, re-

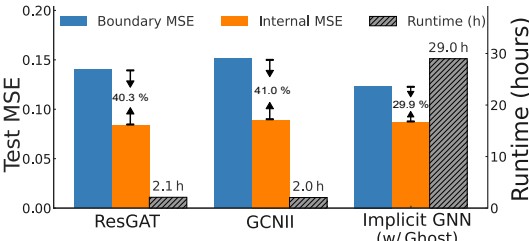

Figure 3: Boundary vs. interior MSE (two left axes) and wall-clock runtime (right axis) for two backbone GNNs and our proposed implicit solver with ghost nodes. Observe that while the implicit solver reduces prediction error at boundary nodes and mitigates the boundary-interior loss gap, it incurs a runtime overhead of over $14\times$ compared to standard GNNs.

spectively, and let $\mathbf{h}_{\mathrm{nbr}_1}$, $\mathbf{h}_{\mathrm{nbr}_2}$ denote the embeddings of the immediate and second-order interior (downstream) neighbors of boundary, respectively. Under a first-order condition, the ghost node proxy is defined as $\mathbf{h}_g = p_{b,g}\mathbf{h}_b + p_{n_1,g}\mathbf{h}_{\mathrm{nbr}_1}$. For second-order, it becomes $\mathbf{h}_g = p_{b,g}\mathbf{h}_b + p_{n_1,g}\mathbf{h}_{\mathrm{nbr}_1} + p_{n_2,g}\mathbf{h}_{\mathrm{nbr}_2}$. Each such augmentation changes the structure of the rows in $\mathbf{A}'$ corresponding to $v_g$, introducing new learnable parameters in $\mathbf{A}'_\Theta$. Upon these intuitions, we define $(i,j)$-th entry of $\mathbf{A}'_\Theta$ as follows.

$$\mathbf{A}'_\Theta[i,j] = \begin{cases} 1 & \text{if } i = j \\ p_{i,j} & \text{if } v_i \in V \text{ and } v_j \in V_{\mathrm{GH}} \\ w_{i,j} & \text{if } v_i \in V \text{ and } v_j \in \mathcal{N}(i) \\ 0 & \text{otherwise} \end{cases}.$$

This $\mathbf{A}'_\Theta$ reduces the implicit solver in Eq. (13) to an explicit solution linear system $\Pi_{\mathbf{A}}\big(\mathbf{A}'_\Theta \mathbf{H}'^{(l+1)}\big) = \mathbf{H}^{(l)}$, which enjoys a closed-form solution $\mathbf{H}'^{(l+1)} = \mathbf{A}'_\Theta{}^\dagger \mathbf{H}'$. Here, $\dagger$ denotes the Moore-Penrose inverse (Prasad & Bapat, 1992), and the ghost node-padded tensor $\mathbf{H}'$ is defined in Eq. (6). We present a concrete examples in App. A.3. One property of the inverse is producing dense matrices (Chamberlain et al., 2021), which reflects the global coupling of the system and allows to bypass the limitation of using a sparse, hand-crafted $\mathbf{A}'$. To operationalize this idea, we learn $\mathbf{A}'_\Theta{}^\dagger$ by approximating the inverse operator through a trainable GNN as

$$\min_{f,\theta_{\mathrm{gh}},\Psi} \ell\big(\mathbf{y}, \ f\big(\Pi_A(\mathbf{H}'^{(l+1)})\big)\big), \tag{14}$$
$$\text{s.t. } \mathbf{H}'^{(l+1)} = \mathbf{A}'_\Theta{}^\dagger \mathbf{H}'^{(l)}, \mathbf{A}'_\Theta{}^\dagger = \Psi(\mathbf{A}'_\Theta), \mathbf{A}'_\Theta{}^\dagger \mathbf{A}'_\Theta \approx \mathbf{I},$$

where the goal is to learn an inverse operator $\Psi$ that approximates $\mathbf{A}'_\Theta{}^\dagger$, and the regularization term $\mathbf{A}'_\Theta{}^\dagger \mathbf{A}'_\Theta \approx \mathbf{I}$ enforces an inverse constraint. We learn this operator because analytical inversion can be computationally expensive and unstable, especially as $\mathbf{A}'_\Theta$ may vary across samples and training iterations. A learned $\Psi$ provides an approximation that generalizes across boundary structures. In implementation, we can parameterize $\Psi$ using a differentiable architecture such as a GNN or low-rank factorization. For the explicit inverse-operator learner, see Algorithm A.10.3 (in App. A.10).

## 4.4 Theoretical Analysis

Theorem 1 shows that our proposed ghost-proxy parameterization can be viewed as a learned regularizer for inverse Robin boundary reconstruction. Specifically, Eq. (4) shows that the ghost state is governed by a

local boundary-closure variable $\eta = (\omega_1, \omega_2)$. In open-boundary settings, $\eta$ is typically unobserved. Directly reconstructing it from noisy prediction data leads to the local inverse problem

$$\widehat{\eta}_{\text{inv}} = \arg\min_{\eta} \frac{1}{2}\|\mathcal{M}(\eta) - y^{\delta}\|_2^2,$$

where $\mathcal{M}(\eta)$ denotes the local forward map that constructs the ghost state from $\eta$ and produces the prediction, and $y^{\delta}$ denotes noisy observations. When this inverse problem is ill-conditioned, direct reconstruction can amplify observation noise.

The ghost module in Eq. (5) provides a learned boundary-closure estimate $\widehat{\eta}_{\theta}(z)$, where $z$ contains the local temporal history of the boundary node and its downstream neighbor, together with graph context. We use $\widehat{\eta}_{\theta}(z)$ as a learned regularization center, yielding the learned-regularized inverse problem

$$\widehat{\eta}_{\lambda} = \arg\min_{\eta} \frac{1}{2}\|\mathcal{M}(\eta) - y^{\delta}\|_2^2 + \frac{\lambda}{2}\|\eta - \widehat{\eta}_{\theta}(z)\|_2^2, \qquad \lambda > 0.$$

**Theorem 1** (Learned ghost regularization reduces local inverse noise amplification)**.** *Let $\eta^{\star}$ denote the target local boundary-closure variable. Assume that $\mathcal{M}$ is differentiable near $\eta^{\star}$, and let $J = \left.\frac{\partial \mathcal{M}}{\partial \eta}\right|_{\eta^{\star}}$ be the local sensitivity matrix from the boundary-closure variable to the prediction. Let $\sigma_k > 0$ be a singular value of $J$. The singular value $\sigma_k$ measures the stability of the corresponding local inverse direction: a smaller $\sigma_k$ indicates a more ill-conditioned direction and stronger noise amplification under direct inversion.*

*Under the local linearized inverse model, direct inverse reconstruction has observation-noise amplification factor $1/\sigma_k$. In contrast, using the learned center $\widehat{\eta}_{\theta}(z)$ for regularization changes the corresponding noise amplification factor to $\sigma_k/(\sigma_k^2 + \lambda)$. Since $\sigma_k/(\sigma_k^2 + \lambda) < 1/\sigma_k$ for any $\lambda > 0$, learned ghost regularization suppresses the noise amplification caused by direct inversion.*

*Proof Sketch.* Linearizing $\mathcal{M}$ near $\eta^{\star}$ gives a least-squares inverse problem governed by the local sensitivity matrix $J$. The unregularized inverse solution amplifies noise by $1/\sigma_k$ along the corresponding singular component. Adding the quadratic regularization centered at $\widehat{\eta}_{\theta}(z)$ changes the singular-value filter factor to $\sigma_k/(\sigma_k^2 + \lambda)$, which damps ill-conditioned inverse directions. We defer the full SVD expansion, error decomposition, and closed-form derivation to App. A.8.

Theorem 1 provides an explanation for the effectiveness of ghost proxies. The ghost node is not merely an additional latent node; it acts as a learned boundary-closure regularizer that stabilizes boundary-context closure used in message passing.

## 5 Experiments

**Datasets.** We evaluate on two directed-network dataset. **(i) River.** A real world river network preprocessed from LamaH-CE2 (Klingler et al., 2021) over the Danube basin, providing hourly discharge and meteorological records. The graph has 358 nodes and 357 directed edges, partitioned into 209 boundary and 149 interior nodes. Each node has five features: discharge, surface air pressure, precipitation, temperature, and soil moisture. **(ii) Blood flow.** A simulated arterial-network dataset generated with *openBF* (Benemerito et al., 2024), on a Circle of Willis model (Vrselja et al., 2014); we use a connected subnetwork with 14 nodes and 14 directed edges. Each node carries four features: flux, pressure, velocity, and cross-sectional area. For both datasets, model inputs are formed by concatenating the past $W$ hours of features along the channel dimension; the prediction target is flux $\mathbf{y} \in \mathbb{R}^{|V|}$ at horizon $t+n$. All variables are independently normalized via z-score to ensure consistent scaling across nodes and variables.

**Metrics.** We evaluate models under a supervised node regression setup, following (Kirschstein & Sun, 2024; Jiang et al., 2025). Given $W$ hours of historical flux data for all nodes, the goal is to predict the flux volume $n$ hours ahead. In our setting, we use $W = 24$ and a forecast horizon of $n = 6$ hours. We compute the mean-squared error (MSE) on three node sets: all nodes $\ell(\hat{\mathbf{y}}, \mathbf{y}) = \frac{1}{|V|}\sum_{v_i \in V}(\hat{y}_i - y_i)^2$, boundary nodes $\ell_{BN}(\hat{\mathbf{y}}, \mathbf{y}) = \frac{1}{|V_{\text{BN}}|}\sum_{v_i \in V_{\text{BN}}}(\hat{y}_i - y_i)^2$, and interior nodes $\ell_{IN}(\hat{\mathbf{y}}, \mathbf{y}) = \frac{1}{|V_{\text{IN}}|}\sum_{v_i \in V_{\text{IN}}}(\hat{y}_i - y_i)^2$.

Table 2: MSE comparison for River (left) and Blood (right), with Boundary/Interior breakdown. Shaded rows show group means: *Avg (Base)* averages the three baselines (GCNII, ResGCN, ResGAT), and *gTFP$_{Avg}$* averages their ghost counterparts. *Diff (%)* is the relative difference between boundary and interior. The rightmost column reports the relative average runtime.

| Flux Predictors | River | | | | Blood | | | | Runtime (Avg., rel.) |
|---|---|---|---|---|---|---|---|---|---|
| | Avg. | Boundary | Interior | Diff. (%) | Avg. | Boundary | Interior | Diff. (%) | |
| GCNII | 0.1255±0.0009 | 0.1510±0.0012 | 0.0893±0.0007 | 40.9±0.8 | 0.0673±0.0005 | 0.1454±0.0011 | 0.0362±0.0003 | 75.1±0.7 | |
| ResGCN | 0.1231±0.0008 | 0.1410±0.0011 | 0.0984±0.0008 | 30.2±0.7 | 0.0570±0.0004 | 0.1138±0.0008 | 0.0342±0.0003 | 70.0±0.6 | ×1.0 |
| ResGAT | 0.1168±0.0011 | 0.1400±0.0010 | 0.0837±0.0006 | 40.2±0.8 | 0.0484±0.0003 | 0.1054±0.0007 | 0.0253±0.0002 | 76.0±0.6 | |
| Avg (Base) | 0.1218±0.0006 | 0.1439±0.0008 | 0.0905±0.0005 | 37.2±0.6 | 0.0576±0.0003 | 0.1215±0.0007 | 0.0319±0.0002 | 73.7±0.5 | |
| gTFP$_{GCNII}$ | 0.1120±0.0008 | 0.1276±0.0010 | 0.0905±0.0007 | 29.1±0.8 | 0.0637±0.0005 | 0.1337±0.0010 | 0.0360±0.0003 | 73.0±0.7 | |
| gTFP$_{ResGCN}$ | 0.1163±0.0008 | 0.1311±0.0009 | 0.0951±0.0007 | 27.8±0.7 | 0.0542±0.0004 | 0.1074±0.0008 | 0.0330±0.0003 | 69.3±0.6 | ×2.6 |
| gTFP$_{ResGAT}$ | 0.1060±0.0007 | 0.1253±0.0009 | 0.0799±0.0006 | 36.2±0.7 | 0.0456±0.0003 | 0.0984±0.0007 | 0.0248±0.0002 | 74.7±0.6 | |
| gTFP$_{Avg}$ | 0.1114±0.0005 | 0.1280±0.0007 | 0.0885±0.0005 | 31.0±0.5 | 0.0545±0.0003 | 0.1132±0.0006 | 0.0313±0.0002 | 72.3±0.5 | |
| Implicit GNN | 0.1117±0.0008 | 0.1275±0.0010 | 0.0901±0.0007 | 29.3±0.7 | 0.0584±0.0004 | 0.0745±0.0007 | 0.0522±0.0004 | 29.8±0.6 | ×13.1 |
| Implicit GNN w/ Ghost | 0.1084±0.0007 | 0.1235±0.0009 | 0.0867±0.0006 | 29.8±0.7 | 0.0559±0.0004 | 0.0707±0.0006 | 0.0498±0.0004 | 29.7±0.6 | ×13.8 |

**Competitors.** We compare with several GNN baselines, including residual variants of graph convolutional networks (ResGCN) (Wu et al., 2019), graph attention networks (ResGAT) (Veličković et al., 2017) and GCNII (Chen et al., 2020). They also serve as backbone for gTFP. We additionally include physics-aware graph models (MP-PDE (Brandstetter et al., 2022) and GNO (Li et al., 2020)) as stronger competitors. We further compare against a dense graph transformation as ablation study (Wang et al., 2025).

**Implementation.** We implement the Robin-type boundary condition to model the structural coupling among $v_g$, $v_b$, and $v_{nbr}$, as defined in Eq. (5). The message-passing layers are implemented using ResGCN, ResGAT and GCNII, with 128 dimensional node embedding, applying **ReLU** activation to every non-linear layer. We adapt the Picard search method (Paniconi & Putti, 1994) to accelerate the implicit GNN training, following Gu et al. (2020). We use a fully connected and trainable adjacency matrix to learn the inverse operator. For ablation study, we learn it over symmetric (*i.e.,* $\mathbf{A}'_{\Theta}{}^{\dagger}$ in Eq. (14)) and asymmetric (*i.e.,* $\mathbf{A}'^{\dagger}\Theta$, $\mathbf{A}'$ in Eq. (6) and $\Theta$ in Eq. (13)) versions, following Wang et al. (2025), with detailed results and analysis deferred to RQ5. For the compared models, we implement their GNN architectures following Kirschstein & Sun (2024) and evaluate their results on both ground-truth, forward flow (*i.e.,* $\mathbf{A}$) and the physically-implausible, reverse flow (*i.e.,* $\mathbf{A}^{\top}$), which enables to verify whether adding boundary condition will improve their topology-awareness during training, with results deferred to RQ2.

**Results and Analysis**

Based on the results in Table 2, 3 and 4, we answer the following research questions (**RQ1–5**):

**RQ1.** *To what extent can ghost nodes mitigate boundary-side errors caused by missing upstream context in GNN training?*

We assess this question by linking our learned boundary-context proxies via Eq. (5) to the boundary nodes of three GNN backbones (ResGAT, ResGCN, GCNII), and measuring their impact on boundary and overall MSE performance. Compared to the MLP baseline, on **River**, adding ghost nodes reduces the boundary MSE by 10.7%, 6.7%, and 15.7%, and reduces overall MSE by 8.83%, 5.68%, 10.53%, for ResGAT, ResGCN, and GCNII, respectively. All three backbones now outperform the MLP baseline (0.1135) on overall MSE by an average of 1.8%. On **Blood**, ghost nodes similarly reduce boundary MSE by 7.01%, 5.96%, and 7.99%, and decrease overall MSE by 5.19%, 4.57%, and 5.34% for ResGAT, ResGCN, and GCNII, respectively. The three backbones outperform the MLP baseline (0.0670) on overall MSE by an average of 17.5%. Averaged across the three GNN backbones, ghost nodes reduce overall MSE by 8.3% and 5.0%, while reducing boundary MSE by 11.0% and 6.9% on River and Blood, respectively. We further show that, although MP-PDE (Brandstetter et al., 2022) and GNO (Li et al., 2020) reduce overall MSE, they still leave a large boundary–interior gap, whereas gTFP achieves larger boundary-side gains (App. A.5.2). Such improvement indicates that adding ghost nodes consistently strengthens GNN performance relative to the MLP baseline, substantiating that learned boundary-context proxies can mitigate errors associated with missing upstream context. We further add paired significance tests on the real River and Blood datasets, showing consistent gains across matched

Table 3: Topological comparison of Fwd, Rev, and `gTFP` on River and Blood. In each cell, the first line shows Avg with its change relative to Rev for the same backbone and dataset; the second line (with results parenthesized) shows Boundary with its change relative to Rev. ↑ indicates better (lower MSE), and ↓ indicates worse (higher MSE).

| Backbone | River | | | Blood | | |
|---|---|---|---|---|---|---|
| | **Fwd** | **Rev** | **gTFP** | **Fwd** | **Rev** | **gTFP** |
| **GCNII** | 0.1255±0.0009 (↑2.7±0.8%) | 0.1288±0.0010 | 0.1120±0.0008 (↑13.1±0.9%) | 0.0673±0.0005 (↓3.5±0.7%) | 0.0651±0.0005 | 0.0637±0.0005 (↑2.0±0.7%) |
| | (0.1510±0.0012, ↓1.5±1.0%) | (0.1489±0.0011) | (0.1276±0.0010, ↑14.3±1.0%) | (0.1454±0.0011, ↓9.3±0.9%) | (0.1331±0.0010) | (0.1337±0.0010, ↓0.5±0.8%) |
| **ResGCN** | 0.1231±0.0008 (↑1.2±0.7%) | 0.1246±0.0009 | 0.1163±0.0008 (↑6.6±0.8%) | 0.0570±0.0004 (↓3.3±0.6%) | 0.0552±0.0004 | 0.0542±0.0004 (↑1.8±0.6%) |
| | (0.1410±0.0011, ↓3.0±0.9%) | (0.1368±0.0010) | (0.1311±0.0009, ↑4.2±0.9%) | (0.1138±0.0008, ↓9.2±0.8%) | (0.1042±0.0008) | (0.1074±0.0008, ↓3.1±0.8%) |
| **ResGAT** | 0.1168±0.0011 (↑0.9±0.8%) | 0.1178±0.0009 | 0.1060±0.0007 (↑10.0±0.9%) | 0.0484±0.0003 (↓4.8±0.6%) | 0.0461±0.0003 | 0.0456±0.0003 (↑1.1±0.6%) |
| | (0.1400±0.0010, ↓3.7±0.9%) | (0.1351±0.0010) | (0.1253±0.0009, ↑7.2±0.9%) | (0.1054±0.0007, ↓9.7±0.8%) | (0.0960±0.0007) | (0.0984±0.0007, ↓2.5±0.7%) |

Table 4: Ablation of inverse-operator variants on River and Blood. Each entry reports the overall MSE (outside parentheses) and the boundary-node MSE (in parentheses). White rows use the directional inverse operator ($\mathbf{A}'^{\dagger}$), and gray rows use the bidirectional inverse operator ($\mathbf{A}'_{\Theta}{}^{\dagger}$).

| Inv. Op. | River | | Blood | |
|---|---|---|---|---|
| | **w/o Ghost** | **w/ Ghost** | **w/o Ghost** | **w/ Ghost** |
| Implicit GNN | 0.1117±0.0008 | 0.1084±0.0007 | 0.0584±0.0004 | 0.0559±0.0004 |
| | (0.1275±0.0010) | (0.1235±0.0009) | (0.0745±0.0007) | (0.0707±0.0006) |
| ResGAT + $\mathbf{A}'^{\dagger}$ | 0.1120±0.0008 | 0.1059±0.0007 | 0.0451±0.0003 | 0.0439±0.0003 |
| | (0.1335±0.0010) | (0.1231±0.0009) | (0.0631±0.0005) | (0.0616±0.0005) |
| ResGAT + $\mathbf{A}'_{\Theta}{}^{\dagger}$ | 0.1161±0.0008 | **0.1035±0.0007** | 0.0450±0.0003 | **0.0433±0.0003** |
| | (0.1357±0.0010) | **(0.1170±0.0008)** | (0.0628±0.0005) | **(0.0609±0.0005)** |
| ResGCN + $\mathbf{A}'^{\dagger}$ | 0.1193±0.0009 | 0.1152±0.0008 | 0.0534±0.0004 | 0.0523±0.0004 |
| | (0.1340±0.0010) | (0.1309±0.0009) | (0.0730±0.0006) | (0.0717±0.0006) |
| ResGCN + $\mathbf{A}'_{\Theta}{}^{\dagger}$ | 0.1196±0.0009 | **0.1128±0.0008** | 0.0530±0.0004 | **0.0517±0.0004** |
| | (0.1376±0.0010) | **(0.1274±0.0009)** | (0.0729±0.0006) | **(0.0707±0.0006)** |
| GCNII + $\mathbf{A}'^{\dagger}$ | 0.1224±0.0009 | 0.1110±0.0008 | 0.0639±0.0005 | 0.0618±0.0004 |
| | (0.1484±0.0011) | (0.1278±0.0009) | (0.0867±0.0007) | (0.0839±0.0007) |
| GCNII + $\mathbf{A}'_{\Theta}{}^{\dagger}$ | 0.1238±0.0009 | **0.1042±0.0007** | 0.0629±0.0005 | **0.0611±0.0004** |
| | (0.1501±0.0011) | **(0.1202±0.0008)** | (0.0852±0.0007) | **(0.0826±0.0007)** |

random seeds with $p < 0.01$ (App. A.9). To validate whether the learned boundary representation aligns with the missing upstream signal itself, we also add controlled known-forcing experiments in App. A.6.

**RQ2.** *Will adding ghost nodes improve the topology-awareness of standard GNNs?*

We evaluate this by comparing the performance of each backbone under forward (true edge direction) and reverse (all edges flipped) graph variants, both before and after adding ghost nodes. We use the relative forward–reverse MSE difference in Table 3 as a metric of topology-awareness.

On **River**, without ghost nodes the forward-reverse gaps for the three GNN backbones are only 1.1%, 1.1%, and 2.9%, indicating minimal sensitivity to edge direction. After adding ghost nodes, these gaps increase substantially to 10.9%, 6.7%, and 15.0%, corresponding to approximately 8.9×, 6.3×, and 4.7× improvement in topology-awareness. On **Blood**, the baselines even *prefer* the reversed graph. We observe that the forward variants perform worse than their reversed counterparts by 4.56%, 3.08%, and 3.69%. With ghost nodes, this trend reverses, and the forward models outperform their counterparts by 0.87%, 1.63%, and 1.85%, demonstrating the gain of directional sensitivity. Averaged across the three backbones, ghost nodes increase the forward-reverse gap on **River** from 1.7% to 10.9%, and on **Blood** they turn a −3.8% forward deficit into a +1.5% forward advantage. As a consistency check, ghost-node gains nearly vanish in the reverse-flow setting in App. A.5.1. These results suggest that ghost nodes improve GNN sensitivity to edge direction and can enhance the topology-awareness of standard GNN architectures.

**RQ3.** *How effectively can the implicit GNN training defined in Eq. (13) learn ghost nodes compared to naïve layer-wise message passing?*

We compare naïve ghost-augmented GNNs against their implicit fixed-point and inverse-operator variants on the **River** and **Blood** benchmarks (Table 4), and evaluate relative MSE reductions for the three GNN backbones. On **River**, the basic implicit solver reduces overall error by 6.9% and 3.5% for $\text{gTFP}_{\text{ResGCN}}$ and $\text{gTFP}_{\text{GCNII}}$ relative to the naïve ghost baselines, while leaving $\text{gTFP}_{\text{ResGAT}}$ unchanged. On **Blood**, it yields an additional 12.7% reduction for $\text{gTFP}_{\text{GCNII}}$. The stronger inverse-operator learner, with a richer connectivity ${\mathbf{A}'_{\boldsymbol{\Theta}}}^{\dagger}$, further reduces errors by 2.8%, 3.0%, and 7.2% on **River** and by 5.0%, 5.0%, and 4.1% on **Blood** across the three backbones.

Averaged across backbones, the basic implicit solver improves over naïve ghost node method by 3.8%, while the inverse-operator learner increases this improvement to 8.3%. These results show that the implicit GNN training learns ghost nodes more effectively than naïve layer-wise message passing, and lead to additional performance gains with the enriched connectivity $\mathbf{A}'_{\boldsymbol{\Theta}}$.

**RQ4.** *Can the explicit inverse operator learning reduce the runtime overhead of implicit computation without compromising prediction accuracy?*

Yes. From Table 4, the explicit inverse-operator learner ($\mathbf{A}'_{\boldsymbol{\Theta}}$) reduces ResGAT's MSE from 0.1082 (Implicit GNN w/ Ghost) to 0.1033 (4.5%) while shrinking the runtime ratio from 13.8 to 6.0, i.e., about 2.3× faster; on the GCNII backbone it lowers the error from 0.1082 to 0.1040 (3.9%) with a similar speed-up. On Blood, explicit inverse-operator learner maintains or improves accuracy, achieving relative improvements of 22.0% and 7.4% on ResGAT and ResGCN, respectively, over the Implicit GNN w/ Ghost reference (0.0557). The runtime acceleration is similar to that observed on River. These results indicate that directly learning the inverse operator achieves higher accuracy with significantly lighter computation than iteratively solving fixed-point equations. We further evaluate sparse explicit inverse-operator variants for a better accuracy–efficiency trade-off in App. A.5.3.

**RQ5.** *Is it better to learn inverse over the parametric adjacency $\mathbf{A}'_{\boldsymbol{\Theta}}$ or its non-parametric $\mathbf{A}'$, why?*

Table 4 shows that, with Ghost nodes enabled, replacing the symmetric adjacency $\mathbf{A}'_{\boldsymbol{\Theta}}$ with its directed counterpart $\mathbf{A}'$ consistently reduces MSE. On **River**, MSE decreases by 2.3%, 2.1%, and 6.2% for ResGAT, ResGCN, and GCNII, respectively. On **Blood**, the same choice yields smaller but still positive gains of 1.36%, 1.53%, and 1.13% for the three backbones.

Although the physical flow graph is strictly downstream-oriented, our bidirectional dense formulation better matches the *global* fixed-point operator implicit GNNs approximate, enabling long-range couplings beyond the observed sparse topology. This agrees with the benefits of dense graph transformations reported by Wang et al. (2025). Physically, both domains are open systems with unobserved exchanges (e.g., rainfall/groundwater/withdrawals in rivers; collateral and micro-circulatory paths in vasculature). Allowing bidirectional edges in the learned adjacency helps absorb these latent inflow–outflow processes, relaxes overly strict conservation biases in the observed graphs, and improves predictive accuracy across datasets.

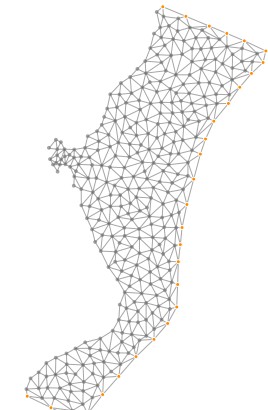

Figure 4: Chesapeake Bay mesh used in RQ6. Orange nodes denote the prescribed.

**RQ6. Do the boundary-dominant error pattern and the gains of gTFP persist on a larger-scale mesh-based bidirectional graph?**

Yes. We find that both the boundary-dominant error pattern and the gains of gTFP persist on a larger-scale mesh-based bidirectional graph. Specifically, this graph is constructed from a Chesapeake Bay hydrodynamic simulation mesh, where nodes correspond to mesh vertices and bidirectional edges follow mesh connectivity. This mesh follows prior unstructured-grid hydrodynamic simulation studies of Chesapeake Bay (Ye et al., 2018). In this setting, reverse edges from interior nodes to boundary nodes exist, so boundary nodes can no longer be identified by zero in-degree. Instead, the boundary is defined geometrically/physically according to where boundary forcing is prescribed or enters the system.

The results remain consistent with the main conclusion of the paper. Despite the different boundary definition, boundary nodes still incur larger prediction errors than interior nodes, and explicitly modeling boundary forcing continues to improve performance. As shown in Figure 4 and Table 5, under ResGAT, the boundary–interior gap remains 7.8%, indicating that boundary nodes remain harder to predict than interior nodes even on this Chesapeake Bay mesh-based bidirectional graph. Adding ghost-node modeling with $\text{gTFP}_{\text{ResGAT}}$ reduces the average MSE from 0.0528 to 0.0515, reduces the boundary MSE from 0.0567 to 0.0545, and shrinks the boundary–interior gap from 7.8% to 6.2%. The stronger variants further improve performance, where implicit GNN w/ Ghost reduces the boundary MSE to 0.0508, and the explicit Inverse-Operator variant further reduces it to 0.0504.

These results show that the boundary-dominant error is not limited to the directed-graph datasets, nor does it rely on the specific zero in-degree boundary definition. Even on a bidirectional graph constructed from a Chesapeake Bay hydrodynamic simulation mesh, where the boundary needs to be specified geometrically/physically, boundary nodes still exhibit larger prediction errors than interior nodes. More importantly, the gains of $\text{gTFP}$ stem from explicitly modeling missing or underrepresented boundary forcing information, rather than from any particular boundary identification rule.

RQ7. Does the explicit inverse-operator provide a better accuracy–efficiency trade-off on the larger mesh-based graph?

Yes. To complement the accuracy results in RQ6, we further evaluate runtime and memory usage on the Chesapeake Bay mesh-based bidirectional graph. As shown in Table 6, adding ghost-node modeling to ResGAT introduces only mild overhead in this setting. Compared with ResGAT, $\text{gTFP}_{\text{ResGAT}}$ increases the training time from 18.1s to 21.2s per epoch and the peak memory from 1.6GB to 1.8GB, while reducing the average MSE from 0.0528 to 0.0515.

| Model | Avg. | Bnd. | Int. | Gap |
|---|---|---|---|---|
| ResGAT | 0.0528 | 0.0567 | 0.0523 | 7.8 |
| $\text{gTFP}_{\text{ResGAT}}$ | 0.0515 | 0.0545 | 0.0511 | 6.2 |
| Implicit GNN | 0.0496 | 0.0519 | 0.0493 | 5.0 |
| Implicit GNN w/ Ghost | 0.0494 | 0.0508 | 0.0492 | 3.1 |
| explicit Inv.-Op. | 0.0492 | 0.0504 | 0.0490 | 2.8 |

Table 5: Corresponding results on the Chesapeake Bay mesh-based bidirectional graph.

By contrast, the implicit fixed-point variants are more expensive. Implicit GNN w/ Ghost requires 42.3s per epoch and 2.9GB peak memory. The explicit Inverse-Operator achieves the best accuracy–efficiency trade-off: it obtains the lowest average MSE of 0.0492, while requiring only 31.4s per epoch and 2.2GB peak memory. Thus, compared with Implicit GNN w/ Ghost, it achieves better accuracy with lower runtime and memory cost. These results support that the explicit Inverse-Operator preserves the boundary-modeling benefit while avoiding the full cost of iterative fixed-point inference on larger graphs.

| Model | Train/Epoch (s) | Peak Mem. (GB) | Avg. MSE |
|---|---|---|---|
| ResGAT | 18.1 | 1.6 | 0.0528 |
| $\text{gTFP}_{\text{ResGAT}}$ | 21.2 | 1.8 | 0.0515 |
| Implicit GNN | 39.8 | 2.7 | 0.0496 |
| Implicit GNN w/ Ghost | 42.3 | 2.9 | 0.0494 |
| explicit Inv.-Op. | 31.4 | 2.2 | 0.0492 |

Table 6: Efficiency comparison on the Chesapeake Bay mesh-based bidirectional graph.

# 6 Related Work

**Graph Augmentation with Virtual Nodes.** Adding virtual nodes to a graph is a known technique to improve expressivity and global information flow (Xu et al., 2019; Ying et al., 2021). A common instantiation is to introduce a single global node connected to all nodes, which has been used to enhance graph-level prediction (Baek et al., 2021), reduce oversquashing by providing shortcut routes for long-range interactions (Hwang et al., 2022), and assist physical simulations where global context or long-range coupling is beneficial (Bentivoglio et al., 2025; Mayr et al., 2023). However, such global augmentation is not tailored to *localized* uncertainty near open boundaries. In our setting, boundary nodes suffer from missing boundary forcing and their errors propagate downstream under directed transport. We therefore introduce *per-boundary* ghost nodes as boundary-specific proxies, explicitly targeting the boundary-context closure deficit. rather than improving global connectivity in a generic manner; the design is motivated by closing local boundary stencils instead of merely increasing model capacity.

**Implicit Graph Neural Networks (IGNNs).** IGNNs compute node embeddings as a fixed point of a nonlinear system (Gu et al., 2020), a paradigm further generalized by deep equilibrium models (Bai et al., 2021; 2020). This line of work provides a principled way to increase effective depth with parameter sharing, and its behavior has been interpreted through numerical diffusion (Chamberlain et al., 2021) and analyzed using monotone operator theory (Baker et al., 2023), with strategies proposed to mitigate oversmoothing and improve stability (Rusch et al., 2023). Despite these advances, a key limitation remains their reliance on iterative fixed-point solvers and convergence constraints, which can be computationally costly on large graphs. Our implicit formulation inherits the equilibrium perspective, while our explicit inverse-operator learner offers an efficient layer-wise alternative that avoids expensive iteration yet preserves boundary-aware coupling. This yields a solver-free forward pass with predictable cost while still capturing the long-range coupling induced by boundary proxies.

**Boundary conditions in PIML.** Boundary conditions (BCs) are central in physics-informed machine learning: prior work balances PDE and BC losses via adaptive weights or architectural constraints, and analyzes optimization and generalization failure modes (Raissi et al., 2019; McClenny & Braga-Neto, 2020; Wang et al., 2023; Krishnapriyan et al., 2021). Operator-learning methods such as FNO often assume periodic BCs, whereas graph-based models better accommodate irregular geometries and mixed/complex BCs (Li et al., 2021; Horie & Mitsume, 2022; Li et al., 2024). Other studies infer unknown BCs or latent forcings from data, typically posed as an inverse problem or a variational formulation (Horuz et al., 2022; Zhao et al., 2022; Frerix et al., 2021).

**Ghost-cell view and inverse perspective.** Classical numerical PDE solvers often introduce *ghost cells/nodes* to close boundary stencils, enabling localized boundary parameterization consistent with the interior discretization (Tseng & Ferziger, 2003). Our approach brings this idea to graph message passing: each boundary node is equipped with a learned ghost proxy, which acts as a learned boundary-context proxy constructed from local boundary and downstream information. In contrast to approaches that fix BCs or enforce them only through auxiliary losses, our ghost-node formulation *jointly* updates boundary-closure variables with interior dynamics in a data-driven framework, and integrates them into both implicit and explicit inference (Liu et al., 2025), enabling consistent boundary–interior coupling.

## 7 Conclusion

This paper revisits the empirical shortcomings of GNNs in topological flux prediction and challenges the prevailing conclusion that GNNs are fundamentally unsuitable for such tasks. By dissecting the prediction loss behavior on fluid networks, we demonstrate that the dominant source of error lies at boundary nodes. To address the boundary-context closure deficit in GNN-based flux prediction, we propose a novel `gTFP` framework, which augments GNNs with ghost nodes and an implicit solver to incorporate physically consistent boundary conditions during training. To improve scalability, we devise an explicit solver that learns inverse operators, enabling efficient layer-wise computation. Experiment demonstrates that `gTFP` improves predictive accuracy and reduces the boundary-interior loss gap across multiple standard GNN backbones. Future work will extend the current boundary-local formulation to spatially distributed external forcing, such as precipitation or wind-driven inputs acting over the interior of the domain.

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

# A   Appendix

## A.1   Different Boundary Conditions

To connect our ghost-node parameterization in Eq. (5) to classical boundary conditions, we instantiate three canonical types on the river graph, namely, Dirichlet, Neumann, and Robin.

**Dirichlet ($u = g_a$).** Yields $u_g = u_b = g_a$. On the graph this degenerates to $h_g^{(l)} = h_b^{(l)}$, i.e., a *fixed* boundary input. This choice does not match the time-varying inflow/pressure in our datasets and therefore performs worst.

**Neumann ($\partial_x u = g_a$).** A first-order discretization gives $u_g = u_b - g_a \, \Delta x$. On the graph, one can use a learned form $h_g^{(l)} = \mathrm{MLP}(h_b^{(l)}; \theta)$, where the effect of $g_a$ is implicitly encoded into the parameters $\theta$.

**Robin.** In Eq. (5), $h_g^{(l)} = \omega_1 \, h_b^{(l)} + \omega_2 \, \frac{h_b^{(l)} - h_{\mathrm{nbr}}^{(l)}}{\Delta x}$, which jointly uses the boundary value and the boundary–downstream difference. We consider two options: (i) a linear two-parameter variant that learns only $\omega_1, \omega_2$; and (ii) the *MLP* variant used in the main paper. Our sensitivity study over MLP depth/width (Table 12) shows low sensitivity.

| Model | Boundary MSE | Interior MSE | Overall MSE |
|---|---|---|---|
| ResGCN | 0.1408 | 0.0985 | 0.1232 |
| Dirichlet | 0.1415 | 0.0990 | 0.1238 |
| Neumann | 0.1406 | 0.0983 | 0.1230 |
| Robin (2-param) | 0.1374 | 0.0974 | 0.1213 |
| Robin (MLP) | 0.1313 | 0.0950 | 0.1162 |

Table 7: Ablation on boundary conditions for ResGCN on the river network.

To connect our ghost-node parameterization in Eq. (5) to classical boundary conditions, we design experiments on three canonical types in the river graph, namely, Dirichlet, Neumann, and Robin. As summarized in Table 7, relative to the vanilla forward ResGCN, Dirichlet-Ghost slightly increases overall MSE by 0.5%, while Neumann-Ghost yields a negligible 0.2% decrease. The simplified two-parameter Robin-Ghost reduces boundary and overall MSE by 2.4% and 1.5%, respectively, and the MLP Robin-Ghost achieves the largest gains, lowering boundary and overall MSE by 6.7% and 5.7%. These quantitative trends support our choice of a Robin-style, learnable ghost mapping as the default.

## A.2   Derivation for $\mathbf{A}'_\Theta$

**Notation (Symbol Table).**   To keep the appendix consistent with the main text, we list the symbols used below. Only the Robin coefficients have been renamed from $(\alpha, \beta)$ to $(w_1, w_2)$; the advection speed $a$ is unchanged, and the spatial step is uniformly denoted by $\Delta x$.

- $u(x, t)$: scalar state; $u_i^n$ is the discrete state at grid index $i$ and time level $n$.

- $a > 0$: advection speed in the governing PDE (kept as is).

- $\Delta x$: spatial grid spacing; $\Delta t$: time step.

- $\sigma := \dfrac{a \, \Delta t}{\Delta x}$: Courant number for advection.

- $w_1, w_2$: Robin boundary coefficients used only in the boundary condition.

- $u_L$: prescribed boundary trace entering the Robin condition; when needed we use $u_L^{n+1}$ to indicate the time level.

- **$\mathbf{A'_{\Theta}}$**: implicit system matrix assembled from interior and boundary discrete equations at time level $n+1$.

This section provides a detailed derivation for the components of the implicit system matrix $\mathbf{A'_{\Theta}}$, establishing the theoretical foundation for the `gTFP` framework discussed in the main paper. By using the 1D advection equation as a canonical example, this derivation serves to:

- Justify the interpretation of GNN message-passing as a numerical discretization of a physical system's spatial dynamics.

- Demonstrate how boundary conditions introduce specific mathematical constraints that are absent in standard GNN formulations.

- Show how these discrete equations naturally form the $\mathbf{A'_{\Theta}}$, which is the core problem our implicit solver addresses.

The derivations are based on two fundamental principles:

- The Governing PDE: The 1D advection equation, $\frac{\partial u}{\partial t} + a\frac{\partial u}{\partial x} = 0$, where $a > 0$.

- The Boundary Condition: Robin-type condition at the boundary $x = 0$, given by $w_1 u + w_2 \frac{\partial u}{\partial x} = u_L$.

### A.2.1 Derivation of the Interior Equations

Here is the derivation of the interior equations:

**Target Equation:**
$$(1 + \sigma)u_i^{n+1} - \sigma u_{i-1}^{n+1} = u_i^n \quad \text{for } i = 1, \ldots, N$$

**Method:** Apply the standard implicit first-order upwind scheme to the governing PDE at interior node $i$.

1. **Discretize the Time Derivative:** We approximate the partial derivative with respect to time, $\frac{\partial u}{\partial t}$, using a first-order forward difference:

$$\frac{\partial u}{\partial t} \approx \frac{u_i^{n+1} - u_i^n}{\Delta t}$$

2. **Discretize the Spatial Derivative (Implicitly):** The term ***implicit*** signifies that the spatial derivative is evaluated at the future time step, $n+1$. The term ***upwind*** (for $a > 0$) means we use a backward difference, looking at the node from which the flow originates $(i - 1)$.

$$\frac{\partial u}{\partial x} \approx \frac{u_i^{n+1} - u_{i-1}^{n+1}}{\Delta x}$$

3. **Combine and Simplify:** Substitute the approximation back into the governing PDE, $\frac{\partial u}{\partial t} + a\frac{\partial u}{\partial x} = 0$:

$$\frac{u_i^{n+1} - u_i^n}{\Delta t} + a\left(\frac{u_i^{n+1} - u_{i-1}^{n+1}}{\Delta x}\right) = 0$$

Multiply the entire equation by $\Delta t$ to clear the denominator:

$$(u_i^{n+1} - u_i^n) + \frac{a\Delta t}{\Delta x}(u_i^{n+1} - u_{i-1}^{n+1}) = 0$$

Let $\sigma = \frac{a\Delta t}{\Delta x}$ be the Courant number. Substituting $\sigma$ gives:

$$u_i^{n+1} - u_i^n + \sigma(u_i^{n+1} - u_{i-1}^{n+1}) = 0$$

4. **Rearrange:** Group all the unknown terms (at time $n + 1$) on the left side and the known terms (at time $n$) on the right side.

$$u_i^{n+1} + \sigma u_i^{n+1} - \sigma u_{i-1}^{n+1} = u_i^n$$

Factoring out $u_i^{n+1}$ yields the final target equation:

$$(1 + \sigma)u_i^{n+1} - \sigma u_{i-1}^{n+1} = u_i^n$$

### A.2.2 Derivation of the Boundary Condition Equation

Here is the Derivation of the Boundary Condition Equation:

**Target Equation:**

$$\left(w_1 - \frac{w_2}{\Delta x}\right)u_0^{n+1} + \left(\frac{w_2}{\Delta x}\right)u_1^{n+1} - u_L^{n+1} = 0$$

**Method:** This equation arises directly from discretizing the Robin boundary condition itself, without involving the PDE.

1. **State the Boundary Condition:** The Robin condition at node $i = 0$ and at the future time step $n + 1$ is:

$$w_1\, u_0^{n+1} + w_2\left.\left(\frac{\partial u}{\partial x}\right)\right|_{i=0}^{n+1} = u_L^{n+1}$$

2. **Discretize the Spatial Derivative:** At the boundary $i = 0$, we cannot use a backward difference. The natural choice is a first-order forward difference, using nodes 0 and 1:

$$\left.\left(\frac{\partial u}{\partial x}\right)\right|_{i=0}^{n+1} \approx \frac{u_1^{n+1} - u_0^{n+1}}{\Delta x}$$

3. **Combine and Rearrange:** Substitute the discretized derivative back into the boundary condition equation:

$$w_1\, u_0^{n+1} + w_2\left(\frac{u_1^{n+1} - u_0^{n+1}}{\Delta x}\right) = u_L^{n+1}$$

Distribute the term $w_2/\Delta x$:

$$w_1\, u_0^{n+1} + \frac{w_2}{\Delta x}u_1^{n+1} - \frac{w_2}{\Delta x}u_0^{n+1} = u_L^{n+1}$$

Group the coefficients for $u_0^{n+1}$ and move all terms to the left side to obtain the final form:

$$\left(w_1 - \frac{w_2}{\Delta x}\right)u_0^{n+1} + \left(\frac{w_2}{\Delta x}\right)u_1^{n+1} - u_L^{n+1} = 0$$

### A.2.3 Derivation of the PDE at the Boundary

Here is the Derivation of the PDE at the Boundary:

**Target Equation:**

$$\left(1 - \frac{a\,\Delta t\,w_1}{w_2}\right)u_0^{n+1} + \left(\frac{a\,\Delta t}{w_2}\right)u_L^{n+1} = u_0^n$$

**Method:** This derivation cleverly combines the PDE and the BC. The key is to use the boundary condition to eliminate the spatial derivative term from the discretized PDE.

1. **Discretize the PDE at the Boundary ($i = 0$):** First, write the implicit discretization of the PDE at node $i = 0$:

$$\frac{u_0^{n+1} - u_0^n}{\Delta t} + a\left.\left(\frac{\partial u}{\partial x}\right)\right|_{i=0}^{n+1} = 0$$

This equation contains the spatial derivative term, which we need to handle.

2. **Isolate the Derivative from the Boundary Condition:** Return to the Robin condition from the previous section:

$$w_1 \, u_0^{n+1} + w_2 \left( \frac{\partial u}{\partial x} \right) \Big|_{i=0}^{n+1} = u_L^{n+1}.$$

We can rearrange this to solve for the derivative term:

$$w_2 \left( \frac{\partial u}{\partial x} \right) \Big|_{i=0}^{n+1} = u_L^{n+1} - w_1 \, u_0^{n+1}$$

$$\left( \frac{\partial u}{\partial x} \right) \Big|_{i=0}^{n+1} = \frac{u_L^{n+1} - w_1 \, u_0^{n+1}}{w_2}$$

3. **Substitute and Simplify:** Now, substitute the expression for the derivative from Step 2 into the discretized PDE from Step 1:

$$\frac{u_0^{n+1} - u_0^n}{\Delta t} + a \left( \frac{u_L^{n+1} - w_1 \, u_0^{n+1}}{w_2} \right) = 0$$

Multiply the entire equation by $\Delta t$:

$$(u_0^{n+1} - u_0^n) + \frac{a\Delta t}{w_2} (u_L^{n+1} - w_1 \, u_0^{n+1}) = 0$$

Distribute the term $\frac{a\Delta t}{w_2}$:

$$u_0^{n+1} - u_0^n + \frac{a\Delta t}{w_2} u_L^{n+1} - \frac{a\Delta t \, w_1}{w_2} u_0^{n+1} = 0$$

4. **Rearrange:** Finally, group the unknown terms $(n+1)$ on the left side and the known term $(n)$ on the right side.

$$\left( 1 - \frac{a\Delta t \, w_1}{w_2} \right) u_0^{n+1} + \left( \frac{a\Delta t}{w_2} \right) u_L^{n+1} = u_0^n$$

In summary, the equations derived for the interior nodes (from the PDE) and the boundary nodes (from the boundary condition) collectively define the rows of the augmented system matrix $\mathbf{A}'_\Theta$. The core contribution of the gTFP framework lies in its ability to learn an efficient, implicit operator.

### A.3 Illustrating the Augmented Matrix

The core of our implicit solver revolves around constructing the system $\mathbf{A}'_\Theta$. The structure of the system matrix $\mathbf{A}'_\Theta$ is a direct reflection of the underlying physical topology. This section provides concrete examples to illustrate this crucial connection. We will demonstrate the structure of $\mathbf{A}'_\Theta$ in complex, coupled ones.

Now, we alter the topology by introducing a new, seventh node, $v_m$, into which both systems merge. The flow paths become: $v_{b1} \to v_{d1} \to v_m$ and $v_{b2} \to v_{d2} \to v_m$.

This introduces physical coupling, as the state of the merge point $v_m$ now depends on inputs from both upstream branches. We augment our state vector with the these node:

$$\mathbf{H}'^{(\mathbf{t+1})} = \left[ \begin{array}{c} \mathbf{h}_{g1}^{(t+1)}, \ \mathbf{h}_{b1}^{(t+1)}, \ \mathbf{h}_{d1}^{(t+1)}, \\ \mathbf{h}_{g2}^{(t+1)}, \ \mathbf{h}_{b2}^{(t+1)}, \ \mathbf{h}_{d2}^{(t+1)}, \ \mathbf{h}_m^{(t+1)} \end{array} \right]^T$$

The new $7 \times 7$ matrix, $\mathbf{A}'_\Theta$, is no longer block-diagonal. The coupling appears precisely in the equation governing the new node $v_m$.

Table 8: Results for the same 14 nodes under three settings (Subgraph / Fullgraph / Ghost-Subgraph). $\%\Delta_{\text{Boundary}}$ denotes the boundary increase relative to Avg, with smaller values indicating a weaker boundary penalty.

| Setting | Avg | Interior | Boundary | $\%\Delta_{\text{Boundary}}$ |
|---------|-----|----------|----------|------------------------------|
| Subgraph | 0.0530 | 0.0355 | 0.0968 | +82.6% |
| Fullgraph | 0.0930 | 0.0891 | 0.1029 | +10.6% |
| Ghost-Subgraph | 0.0476 | 0.0342 | 0.0811 | +70.4% |

$$\mathbf{A}'_{\Theta} = \begin{pmatrix} 1 & -p_{1b} & -p_{1d} & 0 & 0 & 0 & 0 \\ -w_{b1,g1} & 1 & 0 & 0 & 0 & 0 & 0 \\ 0 & -w_{d1,b1} & 1 & 0 & 0 & 0 & 0 \\ 0 & 0 & 0 & 1 & -p_{2b} & -p_{2d} & 0 \\ 0 & 0 & 0 & -w_{b2,g2} & 1 & 0 & 0 \\ 0 & 0 & 0 & 0 & -w_{d2,b2} & 1 & 0 \\ 0 & 0 & -w_{m,d1} & 0 & 0 & -w_{m,d2} & 1 \end{pmatrix}$$

**Analysis of the Coupling:** The first six rows and columns largely retain the decoupled structure. The critical change is in the **last row**, which defines the update for $v_m$. The non-zero elements in columns 3 and 6, $-w_{m,d1}$ and $-w_{m,d2}$, are the mathematical signature of the physical merge. These elements, which reside in the previously zero off-diagonal block area, now link the two subsystems together through the dynamics of $v_m$. This clearly demonstrates how a local change in graph topology induces a specific, predictable change in the global system matrix.

## A.4 From Closed Subgraphs to Open systems

**Problem overview.** The 14 evaluation nodes originate from a larger, 30-node system. Training and evaluating them as an isolated subgraph implicitly imposes a closed-system inductive bias: only intra-graph interactions are modeled, while upstream/downstream exchanges and boundary forcings are omitted. This bias disproportionately affects Boundary nodes, where missing cross-boundary inputs manifest as inflated errors. To mitigate this, we introduce ghost nodes that learn a data-driven proxy for the absent external coupling at the boundary, effectively "half-opening" the subgraph.

**Discussion.** Table 8 shows that when the same 14 nodes are trained and evaluated as a closed subgraph, the boundary penalty relative to the overall average is very high (+82.6%), indicating that the omission of external interactions disproportionately affects boundary behavior. Evaluating the identical nodes inside the full graph substantially reduces this penalty to +10.6%, reflecting the richer upstream/downstream context and constraints available in the larger system. When only the subgraph is available, adding ghost nodes still narrows the gap by learning a data-driven ghost node proxy: the penalty drops from +82.6% to +70.4%, while the overall Avg also improves. Altogether, these results support the view that closed subgraphs amplify boundary difficulty, embedding the subsystem in the larger graph naturally balances errors, and ghost nodes provide a lightweight way to approximate external interactions when access to external nodes is not feasible.

## A.5 Additional Experiments

We report four additional ablations that complement the main results: (i) forward vs. reverse settings of ghost nodes, (ii) physics-aware GNN backbones, (iii) sparse inverse operators, and (iv) sensitivity to the ghost MLP $\theta_{gh}$.

### A.5.1 Reverse-Flow Setting of Ghost Nodes (Ref to RQ2.)

Our central claim concerns the physically correct *forward* setting, where upstream boundary nodes lack incoming information and thus suffer from an boundary-context closure deficit. that ghost nodes are designed to compensate. In the *reverse* setting, the adjacency is flipped, so these nodes already receive inputs from their downstream neighbors; in this regime a ghost node no longer corresponds to a meaningful physical boundary condition. We therefore treat reverse experiments only as a consistency check. As shown in Table 9, on the forward graph adding ghost nodes to ResGCN reduces boundary and overall MSE by 6.7%

and 5.7%, respectively, whereas on the reverse graph it slightly increases boundary and overall MSE by 0.8% and 1.0%. This quantitative asymmetry supports our interpretation that ghost nodes specifically repair the boundary-context closure deficit. rather than acting as a generic over-parameterization.

| Model | Boundary MSE | Interior MSE | Overall MSE |
|---|---|---|---|
| ResGCN (Fwd) | 0.1408 | 0.0985 | 0.1232 |
| ResGCN (Rev) | 0.1369 | 0.1071 | 0.1245 |
| gTFP$_{\text{ResGCN}}$ (Fwd) | 0.1313 | 0.0950 | 0.1162 |
| gTFP$_{\text{ResGCN}}$ (Rev) | 0.1380 | 0.1084 | 0.1257 |

Table 9: Forward-versus-reverse comparison of ResGCN with and without ghost nodes on the river network.

### A.5.2 Physics-Aware GNN Backbones (Ref to RQ1.)

We further ask whether using physics-aware backbones alone can resolve the boundary-context closure deficit.. To this end, we add two strong baselines: a message-passing PDE solvers (**MP PDE**) and a Graph Neural Operator (**GNO**). As shown in Table 10, all physics-aware models improve over vanilla Res-GCN in overall MSE: MP PDE and GNO reduce overall error by 4.5% and 0.7%, respectively, whereas our ghost-based gTFP$_{\text{ResGCN}}$ and Explicit-Inverse Operator (ResGCN) achieve larger reductions of 5.7% and 8.5%.

However, their boundary nodes remain substantially less accurate than interior nodes. Relative to vanilla ResGCN, MP PDE and GNO reduce boundary MSE by only 5.4% and 0.9% and shrink the boundary–interior gap by just 8.8% and 2.1%, respectively, while gTFP$_{\text{ResGCN}}$ and Explicit-Inverse Operator (ResGCN) reduce boundary MSE by 6.7% and 9.4% and reduce the boundary–interior gap by 11.0% and 9.8%, respectively. These quantitative trends support our claim that the main bottleneck lies in the missing boundary forcing, rather than the choice of GNN architecture, and that our ghost module acts as a plug-and-play component that effectively complements physics-aware backbones.

| Model | Boundary MSE | Interior MSE | *Diff. (%)* | Overall MSE |
|---|---|---|---|---|
| ResGCN | 0.1408 | 0.0985 | 42.9 | 0.1232 |
| MP PDE Solver | 0.1332 | 0.0957 | 39.2 | 0.1176 |
| GNO | 0.1395 | 0.0982 | 42.1 | 0.1223 |
| gTFP$_{\text{ResGCN}}$ | 0.1313 | 0.0950 | 38.2 | 0.1162 |
| ResGCN + $\mathbf{A}'_{\Theta}{}^{\dagger}$ | 0.1275 | 0.0919 | 38.7 | 0.1127 |

Table 10: Comparison of ResGCN (Fwd), MP PDE solver, GNO, gTFP$_{\text{ResGCN}}$, and Inverse Operator (Res-GCN) on the river network.

### A.5.3 Sparse Inverse Operator (Ref to RQ3. and RQ4.)

To reduce the $\mathcal{O}(|V|^2 d)$ cost of a fully dense inverse while preserving the global coupling of the explicit solver, we parameterize the inverse $(A'_{\Theta})^{\dagger}$ in a low-rank way:

$$(A'_{\Theta})^{\dagger} \approx S + UV^{\top},$$

where $S$ inherits the sparsity pattern of the original graph, and $U, V \in \mathbb{R}^{|V| \times r}$ with $r \ll |V|$ form a low-rank correction. For a feature matrix $X \in \mathbb{R}^{|V| \times d}$,

$$(S + UV^{\top})X = SX + U(V^{\top}X),$$

which costs $\mathcal{O}(|E|d + 2|V|rd)$ time and stores $\approx |E| + 2|V|r$ parameters, remaining *near-linear* in $|E|$ without ever materializing a dense $|V| \times |V|$ matrix. We keep the inverse-consistency regularizer

$$\mathcal{L}_{\text{reg}} = \lambda \big\| (A'_{\Theta})^{\dagger} A'_{\Theta} - I \big\|_F^2,$$

and a sweep over $\lambda \in \{0, 10^{-4}, 10^{-3}, 10^{-2}\}$ yields at most 2% variation in MSE, indicating low sensitivity.

Table 11 compares dense and sparse inverse variants. Relative to vanilla ResGCN, the dense explicit inverse reduces boundary and overall MSE by 9.4% and 8.5%, respectively, but incurs a $\times 6.0$ runtime overhead. The sparse inverse operators with ranks $r=8$ and $r=32$ further reduce boundary MSE by 21.9% and 22.9% and overall MSE by 25.0% and 25.6%, respectively, while requiring only $\times 2.8$ and $\times 3.1$ runtime. These results show that the sparse, low-rank inverse operator attains the best boundary and overall accuracy with substantially lower cost than the dense explicit operator, offering a favorable accuracy and efficiency trade-off.

| Model | Boundary MSE | Interior MSE | Overall MSE | Runtime (rel.) |
|---|---|---|---|---|
| ResGCN | 0.1408 | 0.0985 | 0.1232 | $\times 1.0$ |
| $\mathtt{gTFP}_{\mathrm{ResGCN}}$ | 0.1313 | 0.0950 | 0.1162 | $\times 2.3$ |
| ResGCN + $\mathbf{A}'_{\Theta}{}^{\dagger}$ | 0.1275 | 0.0919 | 0.1127 | $\times 6.0$ |
| Implicit GNN | 0.1236 | 0.0866 | 0.1082 | $\times 13.8$ |
| Sparse Inv. Op. $(r=8)$ | 0.1099 | 0.0678 | 0.0924 | $\times 2.8$ |
| Sparse Inv. Op. $(r=32)$ | 0.1086 | 0.0677 | 0.0916 | $\times 3.1$ |
| Sparse Inv. Op. $(r=64)$ | 0.1087 | 0.0684 | 0.0919 | $\times 3.3$ |

Table 11: Boundary vs. interior vs. overall MSE on the river network, together with relative runtime.

### A.5.4   Sensitivity to the Ghost MLP $\theta_{gh}$

We examine how sensitive the ghost-node performance is to the capacity of the MLP (ghost node proxy). We vary the depth $L$ and width $H$ of the MLP and report boundary, interior, and average MSE. As shown in Table 12, all four configurations yield very similar errors: relative to the default setting ($L=1$, $H=128$), boundary MSE varies by at most 0.6%, interior MSE by at most 0.7%, and overall MSE by at most 0.6%. This indicates that our method does not rely on a carefully tuned ghost MLP and is robust to the particular choice of $\theta_{gh}$.

| Depth (L) | Width (H) | Boundary MSE | Interior MSE | Overall MSE (Avg) |
|---|---|---|---|---|
| 1 | 64 | 0.1321 | 0.0956 | 0.1169 |
| **1** | **128** | **0.1313** | **0.0950** | **0.1162** |
| 2 | 64 | 0.1306 | 0.0946 | 0.1157 |
| 2 | 128 | 0.1310 | 0.0948 | 0.1160 |

Table 12: Sensitivity of ghost ResGCN (MLP) to the depth and width of the ghost mapping MLP on the river network. All configurations achieve nearly identical boundary and interior errors, indicating low sensitivity to $\theta_{gh}$.

### A.6   Controlled Experiments with Known Upstream Forcing

We add controlled one-dimensional flow experiments to directly test whether the learned ghost representation is aligned with the missing upstream boundary signal. These experiments are designed to separate two claims: improving prediction accuracy and recovering information related to missing external forcing. While prediction improvement alone does not prove forcing recovery on real observational datasets, controlled experiments with known hidden upstream inputs allow us to directly evaluate whether the learned boundary representation compensates for the withheld signal.

**Experimental setup.**   We consider three propagation motifs: Chain, Merge, and Split. In the Chain motif, the full system is $v_0 \to v_1 \to \cdots \to v_N$, where $v_0$ provides the true upstream boundary signal and $v_1$ is the first downstream node in the prediction domain. To mimic an open-boundary forecasting problem, we

simulate the full system but remove $v_0$ from the observed graph, so that the model only observes and predicts on $\{v_1, \ldots, v_N\}$. The Merge motif tests multi-upstream confluence, where two upstream branches merge into a downstream trunk. The Split motif tests downstream bifurcation, where an upstream source feeds a stem that later splits into two downstream branches. Thus, the three motifs cover single-path propagation, multi-source confluence, and branching flow propagation.

In each setting, the model uses 8 historical steps to predict 6-step-ahead flux. We evaluate the boundary error at the first observed node or nodes adjacent to the removed upstream source. Since the true upstream signal is available during data generation but withheld from the model input, these experiments allow us to evaluate whether different boundary treatments compensate for the missing input, rather than only whether they reduce downstream prediction error.

**Metrics.** We report the boundary recovery score

$$\text{Recovery} = \frac{\text{MSE}_{\text{removed}} - \text{MSE}_{\text{method}}}{\text{MSE}_{\text{removed}} - \text{MSE}_{\text{full}}} \times 100\%,$$

where $\text{MSE}_{\text{full}}$ denotes the full-reference system with the true upstream node included, and $\text{MSE}_{\text{removed}}$ denotes the removed-boundary baseline. Thus, 0% corresponds to the removed-boundary baseline and 100% corresponds to the full-reference system.

We also evaluate alignment with the true hidden upstream signal. Specifically, we compute the Pearson correlation between the true upstream signal $s_i$ and the boundary signal decoded from the learned ghost state, denoted by $\hat{s}_i$:

$$\text{Corr.} = \frac{\sum_i (\hat{s}_i - \bar{\hat{s}})(s_i - \bar{s})}{\sqrt{\sum_i (\hat{s}_i - \bar{\hat{s}})^2} \sqrt{\sum_i (s_i - \bar{s})^2}}.$$

| Setting | Params | Boundary MSE ($\times 10^{-3}$) | Recovery | Corr. |
|---|---|---|---|---|
| **Panel A: Chain motif** | | | | |
| Full reference | 17,793 | 0.956 | 100.0% | 1.000 |
| Removed-boundary | 17,793 | 3.924 | 0.0% | – |
| Learned ghost | 20,945 | **1.039** | **97.2%** | **0.887** |
| Global node | 20,945 | 1.749 | 73.3% | 0.251 |
| **Panel B: Merge motif** | | | | |
| Full reference | 17,793 | 0.131 | 100.0% | 1.000 |
| Removed-boundary | 17,793 | 0.508 | 0.0% | – |
| Learned ghost | 20,945 | **0.192** | **83.8%** | **0.830** |
| Global node | 20,945 | 0.271 | 62.9% | 0.752 |
| **Panel C: Split motif** | | | | |
| Full reference | 17,793 | 0.174 | 100.0% | 1.000 |
| Removed-boundary | 17,793 | 0.884 | 0.0% | – |
| Learned ghost | 20,945 | **0.379** | **71.1%** | **0.709** |
| Global node | 20,945 | 0.584 | 42.3% | 0.667 |

Table 13: Controlled one-dimensional flow experiments with known upstream forcing variables.

**Results.** As shown in Table 13, learned ghost nodes achieve the highest boundary recovery across all three motifs: 97.2% on Chain, 83.8% on Merge, and 71.1% on Split. They also obtain the highest correlations with the true hidden upstream signal among the compared non-reference methods: 0.887 on Chain, 0.830 on Merge, and 0.709 on Split.

These results provide direct evidence, in controlled settings with known withheld forcing, that the learned boundary representation is aligned with the missing upstream input. At the same time, on real river, arterial-flow, and Chesapeake Bay datasets, the ghost variables should be interpreted more cautiously as learned boundary-context proxies rather than verified reconstructions of true external forcing.

## A.7 Supplementary Figures

These figures complement the main results by contrasting the message flow topology and efficiency. The bar chart compares Forward and Reverse variants across backbones, showing small gaps between the two, while `gTFP` and the inverse learner consistently reduce test MSE. The scatter plot relates test MSE to runtime, where points closer to the lower-left indicate better accuracy–efficiency trade-offs; `gTFP` occupies this region.

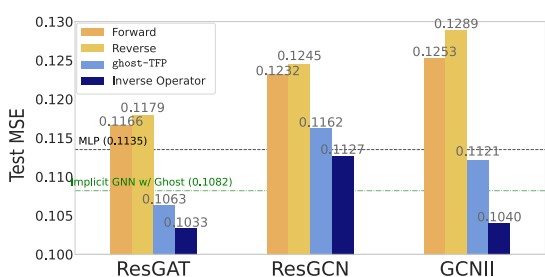

(a) Forward vs. Reverse GNNs (left two bars) highlight the similarity from different message flow direction, while `gTFP` and our inverse learner (right two bars) drive errors lower and widen the gap with reverse flow.

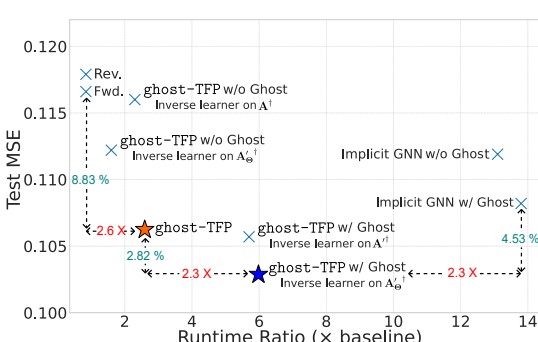

(b) Each point shows model's test MSE against its runtime ratio relative to the ResGAT Forward baseline (1×). Points that fall lower and further left deliver both lower error and faster inference, highlighting superior accuracy–efficiency trade-offs.

## A.8 Derivation of Theorem 1

We provide the derivation for Theorem 1. Recall that Eq. (4) shows that the missing ghost state is governed by the local boundary-closure variable $\eta = (\omega_1, \omega_2)$. In open-boundary forecasting, $\eta$ is not directly observed. Let $\mathcal{M}(\eta)$ denote the local forward map that constructs the ghost state from $\eta$ and produces the prediction. Given noisy prediction data $y^\delta$, direct reconstruction of $\eta$ can be written as

$$\widehat{\eta}_{\text{inv}} = \arg\min_{\eta} \frac{1}{2}\|\mathcal{M}(\eta) - y^\delta\|_2^2.$$

When this inverse problem is ill-conditioned, direct reconstruction can amplify observation noise.

Let $\eta^\star$ denote the target local boundary-closure variable. Assume that $\mathcal{M}$ is differentiable near $\eta^\star$, and let

$$J = \left.\frac{\partial \mathcal{M}}{\partial \eta}\right|_{\eta^\star}$$

be the local sensitivity matrix from the boundary-closure variable to the prediction. A first-order expansion around $\eta^\star$ gives

$$\mathcal{M}(\eta) \approx \mathcal{M}(\eta^\star) + J(\eta - \eta^\star).$$

For noisy prediction data

$$y^\delta = \mathcal{M}(\eta^\star) + \xi,$$

where $\xi$ is observation noise, the corresponding local inverse reconstruction is

$$\widehat{\eta}_\lambda = \arg\min_{\eta} \frac{1}{2}\left\|\mathcal{M}(\eta^\star) + J(\eta - \eta^\star) - y^\delta\right\|_2^2 + \frac{\lambda}{2}\|\eta - \widehat{\eta}_\theta(z)\|_2^2, \qquad \lambda > 0.$$

Here $\widehat{\eta}_\theta(z)$ is the learned boundary-closure estimate induced by the ghost module in Eq. (5), and $z$ contains local temporal history and graph context.

Substituting $y^\delta = \mathcal{M}(\eta^\star) + \xi$, the objective becomes

$$\frac{1}{2}\|J(\eta - \eta^\star) - \xi\|_2^2 + \frac{\lambda}{2}\|\eta - \widehat{\eta}_\theta(z)\|_2^2.$$

This objective is quadratic in $\eta$. The first-order optimality condition is

$$J^\top\big(J(\widehat{\eta}_\lambda - \eta^\star) - \xi\big) + \lambda\big(\widehat{\eta}_\lambda - \widehat{\eta}_\theta(z)\big) = 0.$$

Rearranging gives

$$(J^\top J + \lambda I)\widehat{\eta}_\lambda = J^\top J\eta^\star + J^\top\xi + \lambda\widehat{\eta}_\theta(z).$$

Subtracting $\eta^\star$ from both sides yields the error decomposition

$$\widehat{\eta}_\lambda - \eta^\star = (J^\top J + \lambda I)^{-1}J^\top\xi + \lambda(J^\top J + \lambda I)^{-1}\big(\widehat{\eta}_\theta(z) - \eta^\star\big).$$

The first term is the propagated observation noise, and the second term is the bias induced by the learned regularization center.

Let $J = U\Sigma V^\top$ be the singular value decomposition of $J$, with singular values $\{\sigma_k\}$. For the unregularized direct inverse, a noise component along the left singular vector $u_k$ is mapped to the right singular-vector direction $v_k$ with amplification factor

$$\frac{1}{\sigma_k}.$$

For the learned-regularized inverse, the corresponding amplification factor becomes

$$\frac{\sigma_k}{\sigma_k^2 + \lambda}.$$

Since $\lambda > 0$, for every $\sigma_k > 0$,

$$\frac{\sigma_k}{\sigma_k^2 + \lambda} < \frac{1}{\sigma_k}.$$

Therefore, learned ghost regularization damps the noise amplification caused by direct inverse reconstruction, especially along ill-conditioned components with small $\sigma_k$.

This derivation does not require the ghost module to exactly recover the true physical Robin parameters. Instead, it shows that the learned ghost proxy provides a data-driven regularization center for local boundary-context closure. The total estimation error contains both a damped observation-noise term and a bias term determined by the closeness of $\widehat{\eta}_\theta(z)$ to the target closure variable $\eta^\star$. This supports the proposed ghost-node formulation: the ghost proxy is not merely an additional latent node, but a learned boundary-closure regularizer that uses local temporal history and graph context to stabilize boundary-context information used in message passing.

## A.9 Paired Significance Tests

We report paired significance tests for the main baseline-vs-ghost comparisons in Table 14. For each comparison, we use matched random seeds and apply a two-sided paired $t$-test to the per-seed MSE differences. All comparisons are significant with $p < 0.01$, confirming that the gains are consistent across random seeds.

| Dataset | Paired Comparison | Mean Paired Diff. | $t$-stat | $p$-value |
|---|---|---|---|---|
| River | ResGAT vs. gTFPResGAT | $0.0108 \pm 0.0012$ | 20.68 | $3.23\times10^{-5}$ |
| River | Implicit GNN vs. Ghost-Implicit GNN | $0.0033 \pm 0.0010$ | 7.34 | $1.83\times10^{-3}$ |
| River | Dense Inv.-Op. w/o Ghost vs. w/ Ghost | $0.0121 \pm 0.0012$ | 23.38 | $1.98\times10^{-5}$ |
| Blood | ResGAT vs. gTFPResGAT | $0.0028 \pm 0.0004$ | 14.37 | $1.36\times10^{-4}$ |
| Blood | Implicit GNN vs. Ghost-Implicit GNN | $0.0025 \pm 0.0006$ | 9.57 | $6.65\times10^{-4}$ |
| Blood | Dense Inv.-Op. w/o Ghost vs. w/ Ghost | $0.0019 \pm 0.0004$ | 10.44 | $4.75\times10^{-4}$ |

Table 14: Paired significance tests for the main baseline-vs-ghost comparisons. Mean paired difference is computed as baseline MSE minus ghost-augmented MSE using matched random seeds.

## A.10 Algorithms

Here are the 3 algorithms for the `gTFP` Approach.

---

**Algorithm A.10.1:** Ghost-TFP with Standard GNN Backbone

---

**Initialize** : Graph $G = (V, E)$ and its node features $H$, Set of boundary nodes $V_{BN}$,
Number of GNN layers $L$, Upstream neighbor set of node $v_j$ is $\mathcal{N}(j)$,
Graph augmentation operator $\mathcal{A}$, Ground-truth labels $y$,
Model components: GNN backbone $f_{\text{GNN}}$, Predictor MLP $f_{\text{pred}}$, Ghost MLP,
Loss function $\mathcal{L}$ and Projection operator $\Pi_A$.
**Parameters:** Learnable parameters $\theta_{gb}$ for the Ghost MLP,
$\theta_{\text{GNN}}$ for the GNN backbone, and $\theta_{\text{pred}}$ for the predictor.

$V_{GH}, H_{GH}, E_{GH} \leftarrow \emptyset, \emptyset, \emptyset$;
**for** *each boundary node $v_b \in V_{BN}$* **do**
$\quad$ $v_{nbr} \leftarrow v_{nbr} \in \{v_j \mid v_b \in \mathcal{N}(j)\}$ ; $\qquad\qquad\qquad$ // Get interior neighbor
$\quad$ $h_b, h_{nbr} \leftarrow H[v_b, :], H[v_{nbr}, :]$ ;
$\quad$ $h_g \leftarrow \text{MLP}(\text{Concat}(h_b, h_{nbr}); \theta_{gb})$ ; $\qquad\qquad$ // Learn ghost embedding
$\quad$ Create $v_g$ and add to $V_{GH}$; Append $h_g$ to $H_{GH}$;
$\quad$ Add edge $(v_g, v_b)$ to $E_{GH}$
$H' \leftarrow \text{Concat}(H, H_{GH})$ ; $\qquad\qquad\qquad\qquad\qquad$ // Augmented features
$A' \leftarrow \mathcal{A}(V \cup V_{GH}, E \cup E_{GH})$ ; $\qquad\qquad\qquad$ // Augmented adjacency

$H'^{(L)} \leftarrow f_{\text{GNN}}(A', H'; \theta_{\text{GNN}})$
$H^{(L)} \leftarrow \Pi_A(H'^{(L)})$ ; $\qquad\qquad\qquad\qquad$ // Project back to original nodes
$\hat{y} \leftarrow f_{\text{pred}}(H^{(L)}; \theta_{\text{pred}})$

$\mathcal{L}_{loss} \leftarrow \mathcal{L}(y, \hat{y})$;
Update $\theta_{gb}, \theta_{\text{GNN}}, \theta_{\text{pred}}$ using gradients of $\mathcal{L}_{loss}$;
**return** *Prediction $\hat{y}$*

---

**Algorithm A.10.2:** Implicit Ghost-Boundary Message-Passing

---

**Initialize** : Augmented graph $G' = (A', H')$, Ground truth $y$,
Predictor model $f_{\text{pred}}$, Loss function $\mathcal{L}$.
**Parameters:** Shared weight matrix $\Theta$ with constraint $||\Theta||_\infty \leq 1/\lambda_{pf}(A')$,
Learnable parameters $\theta_{\text{pred}}$ for the predictor.
Max iterations $K$, Convergence tolerance $\epsilon > 0$.

$H'^{(0)} \leftarrow H'$;
**for** $k = 1$ **to** $K$ **do**
$\quad$ $H'^{(k)} \leftarrow \sigma(A' H'^{(k-1)} \Theta)$;
$\quad$ **if** $||H'^{(k)} - H'^{(k-1)}|| < \epsilon$ **then**
$\quad\quad$ break;
$H'^* \leftarrow H'^{(k)}$ ; $\qquad\qquad\qquad\qquad$ // Equilibrium embeddings satisfying Eq. (9)

$H^* \leftarrow \Pi_A(H'^*)$ ; $\qquad\qquad\qquad\qquad$ // Project back to original node set
$\hat{y} \leftarrow f_{\text{pred}}(H^*; \theta_{pred})$;
$\mathcal{L}_{loss} \leftarrow \mathcal{L}(y, \hat{y})$;
$\nabla_{H'^*} \mathcal{L}_{loss} \leftarrow$ Compute gradient from loss up to the equilibrium point;

$g^{(0)} \leftarrow \mathbf{0}$ ; $\qquad\qquad\qquad\qquad\qquad$ // Initialize implicit gradient
$D \leftarrow \sigma'(A' H'^* \Theta)$ ; $\qquad\qquad\qquad$ // Jacobian of activation at equilibrium
**for** $k = 1$ **to** $K$ **do**
$\quad$ $g^{(k)} \leftarrow D \odot ((A')^T g^{(k-1)} \Theta^T + \nabla_{H'^*} \mathcal{L}_{loss})$;
$\quad$ **if** $||g^{(k)} - g^{(k-1)}|| < \epsilon$ **then**
$\quad\quad$ break;
$g^* \leftarrow g^{(k)}$ ; $\qquad\qquad\qquad\qquad\qquad$ // Converged implicit gradient $\nabla_Z l$

Compute $\nabla_\Theta \mathcal{L}_{loss}$ and $\nabla_{\theta_{gh}} \mathcal{L}_{loss}$ using $g^*$ via auto-differentiation;
Update parameters $\Theta, \theta_{gh}, \theta_{pred}$;
**return** *Prediction $\hat{y}$*

---

---

**Algorithm A.10.3:** Explicit Adjacency-Inverse Solver

---

**Initialize** : Augmented graph $G' = (V', E')$, Initial augmented features $H'$,
  Ground truth $y$, Loss function $\mathcal{L}$, Projection operator $\Pi_A$.
**Parameters:** Number of layers $L$, Regularization strength $\lambda$,
  Learnable parameters $\theta_\Psi$ for Inverse operator $\Psi$ ,
  Parametric adjacency $\theta_A$ for $A'_\Theta$ , Predictor $\theta_{pred}$ for $f_{\text{pred}}$ .

---

Construct learnable parametric adjacency ;                                              // upstream $\leftrightarrow$ all downstream
$A'_\Theta$ using parameters $\theta_A$;
$(A'_\Theta)^\dagger \leftarrow \Psi(A'_\Theta; \theta_\Psi)$ ;                          // Approximate inverse with learned operator

$H'^{(0)} \leftarrow H'$;
**for** $l = 0$ **to** $L - 1$ **do**
  $\quad \lfloor \quad H'^{(l+1)} \leftarrow \sigma((A'_\Theta)^\dagger H'^{(l)})$ ;    // Apply GNN layer with non-linearity

$H^{(L)} \leftarrow \Pi_A(H'^{(L)})$ ;                                                   // Project back to original node set
$\hat{y} \leftarrow f_{\text{pred}}(H^{(L)}; \theta_{pred})$;
$\mathcal{L}_{\text{node}} \leftarrow \mathcal{L}(y, \hat{y})$ ;                         // Node prediction loss
$\mathcal{L}_{\text{reg}} \leftarrow \lambda ||(A'_\Theta)^\dagger A'_\Theta - I||_F^2$ ; // Inverse constraint regularization
$\mathcal{L}_{\text{total}} \leftarrow \mathcal{L}_{\text{node}} + \mathcal{L}_{\text{reg}}$;

Update $\theta_\Psi, \theta_A, \theta_{pred}$ using gradients of $\mathcal{L}_{\text{total}}$;
**return** *Prediction $\hat{y}$*

---

