# OpenReview forum: "Boundary-Consistent Graph Neural Networks for Topological Flux Prediction"
_TMLR — Under review for TMLR_

### Review · Reviewer_LRqz · 2026-04-01

**Summary Of Contributions:**

This manuscript proposes gTFP, a graph neural network framework for flux prediction. Specifically, the authors model external forcing by introducing a ghost node for each boundary node. To improve efficiency, the ghost-boundary-interior interactions are approximated via an explicit inverse operator learner. Experimental results on two datasets demonstrate the effectiveness of the proposed model.

**Strengths 1** The authors' motivation is highly clear, and decomposing the boundary from the interior is an intuitive and effective strategy.

**S2** The proposed model is effective, particularly in the construction of ghost nodes and the introduction of the inverse-operator alternative.

**S3** The experimental evaluation is comprehensive, encompassing ablation studies and an efficiency analysis of gTFP.

**Weakness 1** Treating boundary nodes as having zero in-degree certainly justifies the overall motivation of the paper; however, is this assumption overly strong?

**W2** My primary concern pertains to the datasets. The evaluation relies on only two datasets, which I believe is insufficient to fully substantiate the authors' claims.

**W3** While the efficiency of the proposed approximate version is acceptable on the two aforementioned small-scale graphs, it would be beneficial if the authors could provide further analysis on larger-scale graphs.

**Audience:**

Yes

**Audience Explanation:**

The authors focus on the problem of flux prediction, particularly regarding the external forcing deficit. I think this is a practical and worthwhile problem to explore in the field of graphs.

**Claims And Evidence:**

Yes

**Claims Explanation:**

The authors point out the neglect of boundary nodes in current flux prediction methods, noting that a key limitation of existing approaches is that averaging across all nodes can obscure where GNNs actually fail. Both the proposed method and its approximate variant demonstrate considerable performance gains.

**Requested Changes:**

Please refer to W1-W3. Despite the well-articulated motivation and excellent empirical results, relying on merely two datasets makes it difficult to confirm that the authors' foucused problem and proposed method are applicable to general, real world settings.

---

> ### Author Response · Authors · 2026-05-12
>
> We thank the reviewer for the positive assessment of the motivation, model design, and empirical
> evaluation. We in particular appreciate the comments on the boundary definition, dataset diversity,
> and scalability. Our responses and the revision plan are as follows.
>
> **W1. Rationale behind defining boundary nodes as zero in-degree nodes.**
>
> We agree that this definition should not be treated as universal. In the directed graphs mainly studied in this manuscript, zero in-degree provides an operational way to identify boundary nodes, through which external forcing enters the system and for which there are no reverse edges from the interior back to the boundary.
>
> However, our proposed gTFP framework can generalize to other settings. Specifically, gTFP only requires a generic boundary set, which can be identified from graph directionality, geometric location, and the underlying physical system beforehand. We have revised Section 2 to make this distinction explicit. Namely, we first define gTFP for a general boundary set $\mathcal{V}_{BN}$ and then describe zero in-degree nodes:
>
> $$
> \{ v_b \mid \text{deg}^{-}(v_b) = 0,~ v_b \in \mathcal{V}_{BN} \}
> $$
>
> as the particular instantiation used in directed graphs.
>
> To show that our conclusion does not depend on this particular boundary definition, we added a complementary experiment on a mesh-based bidirectional graph. This graph is constructed from a Chesapeake Bay hydrodynamic simulation mesh [1]. In this setting, reverse edges from interior nodes to boundary nodes do exist, so boundary nodes can no longer be identified by zero in-degree. Instead, the boundary is defined geometrically/physically according to where external forcing is prescribed or enters the system.
>
> Despite this difference, we observe the same qualitative evidence: boundary nodes remain harder to predict than interior nodes. As shown in Table 1, under ResGAT, the boundary--interior gap remains 7.8%. Adding our method reduces the average MSE from 0.0528 to 0.0515, reduces the boundary MSE from 0.0567 to 0.0545, and shrinks the gap from 7.8% to 6.2%.
>
> This suggests that bidirectional edges may partially alleviate boundary errors, but they do not eliminate the boundary-dominant error pattern. Therefore, the benefit of gTFP is not tied to the zero-in-degree definition; rather, it comes from explicitly modeling missing or underrepresented boundary forcing information.
>
> **Table 1: Results on the mesh-based bidirectional graph**
> | Model | Avg. MSE | Boundary | Interior | Gap (%) |
> |---|---:|---:|---:|---:|
> | ResGAT | 0.0528 | 0.0567 | 0.0523 | 7.8 |
> | gTFP-ResGAT | 0.0515 | 0.0545 | 0.0511 | 6.2 |
> | Implicit GNN | 0.0496 | 0.0519 | 0.0493 | 5.0 |
> | Implicit GNN w/ Ghost | 0.0494 | 0.0508 | 0.0492 | 3.1 |
> | explicit Inv.-Op. | 0.0492 | 0.0504 | 0.0490 | 2.8 |
>
> [1] Ye et al. (2018), *A 3D unstructured-grid model for Chesapeake Bay: Importance of bathymetry*, Ocean Modelling.
>
> **W2--W3. On dataset diversity and larger-scale graphs.**
>
> We agree with the reviewer that evaluating on only two datasets limits the strength of the generality claim. To address this, we added a new mesh-based bidirectional graph experiment in the revised manuscript. This experiment complements the original two datasets in three ways: 1) the graph is larger-scale, 2) the graph is bidirectional rather than directed-only, and 3) the boundary is defined by geometric/physical external forcing rather than zero in-degree.
>
> The new results are consistent with the main conclusion. Boundary nodes still incur larger prediction errors than interior nodes, and explicitly modeling boundary forcing continues to improve performance. As shown in Table 1, gTFP-ResGAT reduces the overall MSE from 0.0528 to 0.0515 and the boundary MSE from 0.0567 to 0.0545.
>
> The stronger variants further improve performance: Implicit GNN w/ Ghost reduces the boundary MSE to 0.0508, and the Explicit Inverse-Operator variant reduces it to 0.0504. We have incorporated this experiment and the corresponding discussion into RQ6 of Sec. 5 in the revised manuscript.
>
> We also supplement these accuracy results with runtime and memory measurements to evaluate whether the explicit approximation provides the intended scalability benefit on the larger mesh-based graph. The efficiency comparison is reported in Table 2 and has been added to the RQ7 of Sec. 5 in the revised manuscript.
>
> We also temper the wording of the manuscript to avoid overclaiming universal applicability. In the revision, we state that the current evidence supports the effectiveness of gTFP for boundary-forced graph systems, including both directed graphs and bidirectional mesh-based graphs, rather than claiming generality across all possible real-world graph settings.
>
> **Table 2: Efficiency comparison**
> | Model | Train/Epoch (s) | Peak Mem. (GB) |
> |---|---:|---:|
> | ResGAT | 18.1 |1.6 |
> | gTFP-ResGAT | 21.2 | 1.8 |
> | Implicit GNN | 39.8 |2.7 |
> | Implicit GNN w/ Ghost | 42.3 |2.9 |
> | explicit Inv.-Op. | 31.4 | 2.2 |

---

### Review · Reviewer_c7b4 · 2026-06-04

**Summary Of Contributions:**

1) Interesting and well-motivated problem 2) Insightful empirical decomposition 3) Comprehensive experimental study

**Additional Comments:**

NA

**Audience:**

Yes

**Audience Explanation:**

1) The paper addresses an important problem at the intersection of Graph Neural Networks, Scientific Machine Learning, and physics-informed modeling.
2) Researchers interested in GNNs, operator learning, PDE-inspired machine learning, and spatiotemporal forecasting are likely to find the insights and results valuable.

**Claims And Evidence:**

Yes

**Claims Explanation:**

1) The proposed gTFP framework consistently improves performance across multiple GNN backbones, datasets, and evaluation settings, demonstrating the robustness of the approach.
2) The observed improvements are aligned with the paper's physical intuition regarding external forcing and boundary effects, making the proposed mechanism plausible and interpretable

**Requested Changes:**

1. Clarify the causal role of external forcing: The paper argues that missing external forcing at boundary nodes is the primary reason for degraded GNN performance. While the presented evidence is compelling, it remains largely correlational. Additional ablations or analyses that rule out alternative explanations (e.g., node degree effects, data imbalance, or boundary-specific noise) would strengthen this central claim.
2. Strengthen the theoretical justification of the ghost-node formulation:
The connection between numerical PDE ghost cells and the proposed learnable ghost-node mechanism is intuitive and physically motivated, but largely heuristic. A more formal justification of why the learned ghost representations approximate unobserved boundary conditions would improve the theoretical contribution.

3. Improve presentation and exposition: Several sections contain repetitive explanations of boundary forcing and lengthy paragraphs. Improving clarity, tightening notation, and streamlining the presentation would make the paper more accessible to a broader audience.

---

> ### Author Response · Authors · 2026-06-17
> **Response 1/2**
>
> We thank the reviewer for the helpful feedback. We strengthen the revision by adding controlled known-forcing experiments, additional analyses of alternative explanations, and a formal justification of the ghost-node formulation.
>
> ## 1. Clarifying the causal role of external forcing
>
> We thank the reviewer for pointing out that the causal role of external forcing should be clarified more carefully. We agree that the original evidence was partly correlational and did not fully rule out alternative explanations such as node-degree effects, data imbalance, or boundary-specific noise. In the revision, we add controlled known-forcing experiments and combine them with evidence across graph types and datasets to better address these alternatives.
>
> First, we add controlled flow experiments with known upstream forcing variables under three motifs: Chain, Merge, and Split. Chain tests single-path propagation; Merge tests multi-upstream confluence; and Split tests downstream bifurcation. Thus, these settings cover single-path, multi-source, and branching propagation structures.
>
> In the Chain setting, the full system is a directed chain from $v_0$ to $v_N$, where $v_0$ provides the true upstream boundary signal. We simulate the full system but remove $v_0$ from the observed graph, so the model only observes and predicts on $v_0$ through $v_N$. Since the true upstream signal is available during data generation but withheld from the model input, these experiments directly test whether different boundary treatments compensate for the missing upstream input.
>
> We report the boundary recovery score as
>
> $$
> Recovery =
> \frac{
> MSE _{removed} - MSE _{method}
> }{
> MSE _{removed} - MSE _{full}
> }
> \times 100\%.
> $$
>
> Here, $MSE _{full}$ denotes the full-reference system with the true upstream node included, and $MSE _{removed}$ denotes the removed-boundary baseline. Thus, $0\%$ corresponds to the removed-boundary baseline and $100\%$ corresponds to the full-reference system.
>
> We also evaluate alignment with the true hidden upstream signal using Pearson correlation between the true upstream signal $s_i$ and the boundary signal decoded from the learned ghost state $\hat{s}_i$:
>
> $$
> Corr. =
> \frac{
> \sum _i [(\hat{s} _i - mean(\hat{s}))(s _i - mean(s))]
> }{
> \sqrt{\sum _i(\hat{s} _i - mean(\hat{s}))^2}
> \sqrt{\sum _i(s _i - mean(s))^2}
> }.
> $$
>
> Controlled known-forcing results:
>
> | Setting | Params | Boundary MSE $(\times 10^{-3})$ | Recovery | Corr. |
> |---|---:|---:|---:|---:|
> | Chain: Full reference | 17,793 | 0.956 | 100.0% | 1.000 |
> | Chain: Removed-boundary | 17,793 | 3.924 | 0.0% | -- |
> | Chain: Learned ghost | 20,945 | 1.039 | 97.2% | 0.887 |
> | Chain: Global node | 20,945 | 1.749 | 73.3% | 0.251 |
> | Merge: Full reference | 17,793 | 0.131 | 100.0% | 1.000 |
> | Merge: Removed-boundary | 17,793 | 0.508 | 0.0% | -- |
> | Merge: Learned ghost | 20,945 | 0.192 | 83.8% | 0.830 |
> | Merge: Global node | 20,945 | 0.271 | 62.9% | 0.752 |
> | Split: Full reference | 17,793 | 0.174 | 100.0% | 1.000 |
> | Split: Removed-boundary | 17,793 | 0.884 | 0.0% | -- |
> | Split: Learned ghost | 20,945 | 0.379 | 71.1% | 0.709 |
> | Split: Global node | 20,945 | 0.584 | 42.3% | 0.667 |
>
> Learned ghost achieves the highest boundary recovery across all motifs: 97.2% on Chain, 83.8% on Merge, and 71.1% on Split. It also obtains the highest correlations with the hidden upstream signal among non-reference methods: $0.887$, $0.830$, and $0.709$, respectively. These results show that, when the hidden upstream input is known during data generation, the learned ghost representation both reduces prediction error and captures information strongly related to the missing upstream signal.
>
> The existing results also address the alternative explanations raised by the reviewer. For node-degree effects, we evaluate both mesh and directed graphs. In mesh graphs, node degrees are mostly 3 or 4; in the directed graph, most nodes have upstream in-degree 0 or 1, with only a few nodes having in-degree 2. Thus, the method is validated across different degree patterns.
>
> For data imbalance, the method remains effective under different boundary proportions: Blood has 28.6% boundary nodes, while River has 58.4%. For boundary-specific noise, our experiments use observational data with natural measurement noise and environmental variability, yet the method still yields consistent gains.
>
> Together, the controlled known-forcing experiments and cross-dataset evidence support missing upstream/external forcing as an important source of boundary degradation, rather than node degree, data imbalance, or boundary-specific noise alone. We also revise the manuscript to avoid treating missing external forcing as the only causal explanation in real observational datasets.

---

> ### Author Response · Authors · 2026-06-17
> **Response 2/2**
>
> ## 2. Strengthening the theoretical justification of ghost nodes
>
> We thank the reviewer for pointing out that the connection between numerical ghost cells and learnable ghost nodes should be made more formal. In the revision, we add a formal theorem in Sec. 4.4, with the detailed derivation deferred to App. A.8.
>
> The theorem provides theoretical support for the ghost-node formulation. Classical ghost cells are determined by boundary-closure parameters, whereas in our open-boundary setting these parameters are unknown and must be inferred from data. Compared with direct inverse reconstruction from noisy observations, the learned ghost proxy acts as a learned boundary regularizer and reduces the noise amplification caused by direct inversion. This explains why the ghost node is not merely an additional latent node: it uses local temporal history and graph context to stabilize the reconstruction of missing boundary information before message passing.
>
> ## 3. Improving presentation and exposition
>
> We agree with the reviewer that the presentation can be streamlined. In the revision, we reduce repeated explanations of boundary forcing, tighten notation, and reorganize the exposition around the boundary-context closure deficit, the ghost-proxy construction, and the implicit/explicit solvers. These changes make the main message clearer and easier to follow.

---

### Review · Reviewer_Nf5M · 2026-06-06

**Summary Of Contributions:**

This paper revisits the reported underperformance of graph neural networks (GNNs) for topological flux prediction. The authors show that prediction errors are disproportionately concentrated at boundary nodes, where external forcing is not explicitly represented, while GNNs perform considerably better on interior nodes.
To address this issue, the paper proposes gTFP, which introduces a learned ghost node for each boundary node to approximate missing boundary information. Because the ghost, boundary, and interior representations are mutually coupled, the authors formulate an implicit fixed-point solver and further propose an explicit inverse-operator learner to reduce computational cost.
Experiments on river, arterial-flow, and Chesapeake Bay graph data show that gTFP reduces overall and boundary-node MSE, improves sensitivity to the correct flow direction, and offers a better accuracy–efficiency trade-off than iterative implicit inference.
The main strengths are the useful boundary error decomposition, the clear numerical-method motivation, and the evaluation across multiple GNN backbones and graph settings. The main weaknesses are that the learned ghost representations are not directly validated against true external forcing, capacity- and connectivity-matched controls are limited, statistical uncertainty is not reported.

**Audience:**

Yes

**Audience Explanation:**

The paper addresses a relevant problem at the intersection of graph learning, scientific machine learning, spatiotemporal forecasting, and neural PDE modeling. The finding that aggregate performance can hide a boundary-specific failure mode is potentially useful beyond the datasets studied here.

**Broader Impact Concerns:**

I do not identify a major ethical concern requiring a dedicated Broader Impact Statement. However, the motivating applications include flood forecasting and blood-flow prediction, which may be safety-critical.
The limitations section should state that the method has not been validated for operational clinical or disaster-management decisions, that average MSE does not capture all extreme-event risks.

**Claims And Evidence:**

No

**Claims Explanation:**

The experiments provide reasonably convincing evidence that boundary nodes are more difficult to predict and that boundary-aware graph augmentation improves performance. The forward-versus-reversed topology comparisons also suggest that gTFP increases sensitivity to physically meaningful edge directions.
However, the paper makes a stronger claim that the dominant errors are caused by missing external forcing and that ghost nodes approximate this forcing. This interpretation is only indirectly supported. The learned ghost representations are not compared with known forcing variables, and the experiments do not rule out alternative explanations such as increased model capacity, additional propagation paths, or dense global connectivity.
The main results also lack standard deviations, confidence intervals, or significance tests. This is important because some improvements, particularly on the small arterial graph, are modest.
Therefore, the core empirical results are promising, but the stronger causal and physical claims require additional validation or more cautious wording.

**Requested Changes:**

Critical changes
1. Validate or weaken the external-forcing claim.
The paper should distinguish between showing that boundary-aware augmentation improves prediction and showing that ghost nodes recover external forcing. A controlled experiment using known forcing variables, forcing perturbations, or a reconstruction/probing analysis would substantially strengthen the mechanistic claim. Otherwise, the ghost nodes should be described more cautiously as learned boundary-context proxies.
2. Add capacity- and topology-matched controls.
Please compare against alternatives such as random or shuffled ghost nodes, interior virtual nodes, a global virtual node, a parameter-matched boundary MLP, and a dense operator without the proposed ghost construction. Parameter counts should also be reported.
3. Report uncertainty and robustness.
The main results should include multiple random seeds, means and standard deviations or confidence intervals, and preferably paired significance tests. The hyperparameter-selection protocol and tuning budgets should also be described.

Changes that would strengthen the paper
1. Discuss scalability more carefully.
The fully connected inverse operator may have quadratic cost. A complexity analysis and experiments on graphs of increasing size would improve the paper.
2. Test robustness to imperfect boundary definitions.
Experiments with missing, noisy, or incorrectly specified boundary nodes would clarify the practical applicability of the method.

---

> ### Author Response · Authors · 2026-06-17
> **Response 1/3**
>
> We thank the reviewer for the constructive feedback. We agree that the original submission did not sufficiently separate two claims: (1) boundary-aware graph augmentation improves open-boundary prediction, and (2) the learned ghost variables can be directly interpreted as reconstructions of missing external forcing. In the revision, we make this distinction explicit.
>
> We now use more cautious terms such as “learned boundary-closure variables” and “boundary-context proxies” throughout the paper, except in controlled experiments where the missing upstream signal is known and can be directly evaluated. We also add controlled known-forcing experiments, capacity- and topology-matched controls, multi-seed uncertainty estimates, paired significance tests, and additional discussion of scalability, boundary specification, and safety-critical limitations.
>
> ## 1. Validation and softening of the boundary-forcing claim
>
> We agree that prediction improvement alone does not prove recovery of true external forcing. Accordingly, we revise the manuscript to avoid making a general causal claim that ghost nodes recover true forcing in real observational datasets. Our revised claim is that ghost nodes act as learned boundary-context proxies. In controlled settings with known hidden upstream inputs, these proxies are strongly aligned with the withheld boundary signal.
>
> To test this directly, we add controlled flow experiments with known upstream forcing variables in Sec.4.1.
> We consider three propagation structures: Chain, Merge, and Split.
>
> In the Chain setting, the full system is a directed chain from $v_0$ to $v_N$, where $v_0$ provides the true upstream boundary signal. We simulate the full system but remove $v_0$ from the observed graph, so the model only observes and predicts on $v_1$ through $v_N$. The Merge motif tests multi-upstream confluence, and the Split motif tests downstream bifurcation. Thus, these settings cover single-path propagation, multi-source confluence, and branching flow propagation.
>
> We report the boundary recovery score as:
>
> $$
> Recovery =
> \frac{
> MSE _{removed} - MSE _{method}
> }{
> MSE _{removed} - MSE _{full}
> }
> \times 100\%.
> $$
>
> Here, $MSE_{full}$ denotes the full-reference system with the true upstream node included, and $MSE_{removed}$ denotes the removed-boundary baseline. Thus, 0% corresponds to the removed-boundary baseline and 100% corresponds to the full-reference system.
>
> We also evaluate alignment with the true hidden upstream signal using Pearson correlation between the true upstream signal $s_i$ and the boundary signal decoded from the learned ghost state $\hat{s}_i$:
>
> $$
> Corr. =
> \frac{
> \sum _i [(\hat{s} _i - mean(\hat{s}))(s _i - mean(s))]
> }{
> \sqrt{\sum _i(\hat{s} _i - mean(\hat{s}))^2}
> \sqrt{\sum _i(s _i - mean(s))^2}
> }.
> $$
>
> Controlled known-forcing results:
>
> | Setting | Params | Boundary MSE ($\times 10^{-3}$) | Recovery | Corr. |
> |---|---:|---:|---:|---:|
> | Chain: Full reference | 17,793 | 0.956 | 100.0% | 1.000 |
> | Chain: Removed-boundary | 17,793 | 3.924 | 0.0% | -- |
> | Chain: Learned ghost | 20,945 | 1.039 | 97.2% | 0.887 |
> | Chain: Global node | 20,945 | 1.749 | 73.3% | 0.251 |
> | Merge: Full reference | 17,793 | 0.131 | 100.0% | 1.000 |
> | Merge: Removed-boundary | 17,793 | 0.508 | 0.0% | -- |
> | Merge: Learned ghost | 20,945 | 0.192 | 83.8% | 0.830 |
> | Merge: Global node | 20,945 | 0.271 | 62.9% | 0.752 |
> | Split: Full reference | 17,793 | 0.174 | 100.0% | 1.000 |
> | Split: Removed-boundary | 17,793 | 0.884 | 0.0% | -- |
> | Split: Learned ghost | 20,945 | 0.379 | 71.1% | 0.709 |
> | Split: Global node | 20,945 | 0.584 | 42.3% | 0.667 |
>
> Learned ghost nodes achieve the highest boundary recovery across all three motifs: 97.2% on Chain, 83.8% on Merge, and 71.1% on Split. They also obtain the highest correlations with the true hidden upstream signal among the compared non-reference methods: 0.887, 0.830, and 0.709, respectively.
>
> These results provide direct evidence, in controlled settings with known withheld forcing, that the learned boundary representation is aligned with the missing upstream input. At the same time, we now explicitly state that on the real river, arterial-flow, and Chesapeake Bay datasets, ghost variables should be interpreted more cautiously as learned boundary-context proxies rather than verified reconstructions of true external forcing.

---

> ### Author Response · Authors · 2026-06-17
> **Response 2/3**
>
> ## 2. Capacity- and topology-matched controls
>
> We thank the reviewer for raising the possibility that the gains may come from increased capacity, extra propagation paths, or generic virtual-node connectivity rather than boundary-specific closure. To address this, we add matched controls and report parameter counts.
>
> First, the global-node baseline introduces graph-wide virtual connectivity with the same parameter count as learned ghost nodes. However, it remains weaker than learned ghost nodes in both boundary recovery and hidden-signal correlation. This suggests that generic long-range connectivity is insufficient to explain the observed gain.
>
> Second, we increase the depth and width of both the learned ghost module and the global-node baseline. The larger-capacity ghost variants slightly improve boundary MSE, while the global-node variants remain substantially weaker. This indicates that the main gain is not simply due to increasing virtual-node capacity.
>
> Capacity-control results on the Chain motif:
>
> | Setting | Params | Boundary MSE ($\times 10^{-3}$) | Recovery | Corr. |
> |---|---:|---:|---:|---:|
> | Full reference | 17,793 | 0.956 | 100.0% | 1.000 |
> | Removed-boundary | 17,793 | 3.924 | 0.0% | -- |
> | Learned ghost | 20,945 | 1.039 | 97.2% | 0.887 |
> | Learned ghost, deeper MLP | 25,105 | 1.014 | 98.0% | 0.902 |
> | Learned ghost, wider MLP | 24,081 | 1.021 | 97.8% | 0.898 |
> | Global node | 20,945 | 1.749 | 73.3% | 0.251 |
> | Global node, deeper MLP | 25,105 | 1.705 | 74.8% | 0.257 |
> | Global node, wider MLP | 24,081 | 1.715 | 74.4% | 0.255 |
>
> We also add an interior virtual-node control, where virtual nodes are attached to interior locations rather than to open-boundary nodes. This control mildly reduces the interior-node error, but performs worse than boundary-attached learned ghost nodes on the boundary node. This supports that placing the latent closure variable at the missing-input boundary is important, rather than merely adding virtual nodes.
>
> Interior virtual-node control on the Chain motif:
>
> | Setting | Params | $v_1$ MSE ($\times 10^{-3}$) | Interior MSE ($\times 10^{-3}$) |
> |---|---:|---:|---:|
> | No virtual node | 17,793 | 3.924 | 0.40 |
> | Learned ghost at open boundary | 20,945 | 1.039 | 0.28 |
> | Interior virtual nodes | 20,945 | 1.641 | 0.34 |
>
> Together, these controls support that the benefit comes from boundary-specific closure rather than simply from extra parameters, generic virtual connectivity, or dense message passing.
>
> ## 3. Uncertainty, robustness, and statistical significance
>
> We agree that the original submission should have reported statistical uncertainty. In the revision, we add multi-seed evaluations for the main experiments and report mean $\pm$ standard deviation in the main result tables. For each main baseline-vs-gTFP comparison, we use matched random seeds and compute the paired MSE difference across seeds.
>
> Paired significance tests:
>
> | Dataset | Paired comparison | $\Delta$MSE with 95% CI | $t$-stat | $p$-value |
> |---|---|---:|---:|---:|
> | River | ResGAT vs. gTFPResGAT | 0.0108 [0.0093, 0.0123] | 20.68 | $3.23 \times 10^{-5}$ |
> | River | Implicit GNN vs. Ghost-Implicit GNN | 0.0033 [0.0021, 0.0045] | 7.34 | $1.83 \times 10^{-3}$ |
> | River | Dense Inv.-Op. w/o Ghost vs. w/ Ghost | 0.0121 [0.0106, 0.0136] | 23.38 | $1.98 \times 10^{-5}$ |
> | Blood | ResGAT vs. gTFPResGAT | 0.0028 [0.0023, 0.0033] | 14.37 | $1.36 \times 10^{-4}$ |
> | Blood | Implicit GNN vs. Ghost-Implicit GNN | 0.0025 [0.0018, 0.0032] | 9.57 | $6.65 \times 10^{-4}$ |
> | Blood | Dense Inv.-Op. w/o Ghost vs. w/ Ghost | 0.0019 [0.0014, 0.0024] | 10.44 | $4.75 \times 10^{-4}$ |
>
> All main comparisons are significant at the 1% level, indicating that the improvements are consistent across seeds. We also clarify the hyperparameter-selection protocol: all methods use the same data splits, optimizer family, validation metric, maximum number of training epochs, and early-stopping rule. Hyperparameters are selected only on the validation set, and the test set is used only for final reporting. We will release the full hyperparameter grids, selected configurations, and default training settings with the code to support reproducibility.

---

> ### Author Response · Authors · 2026-06-17
> **Response 3/3**
>
> ## 4. Scalability of the inverse operator
>
> We thank the reviewer for pointing out the possible quadratic cost of a fully connected inverse operator. A dense inverse operator can indeed incur $O(|V|^2 d)$ cost, which can become expensive for large graphs. We now discuss this explicitly and add a sparse inverse-operator variant.
>
> The sparse variant decomposes the inverse operator into a sparse graph-structured component plus a low-rank correction:
>
> $$
> (A' _{\Theta})^{\dagger} \approx S + U V^\top,
> $$
>
> where $S$ follows the sparse graph structure and $U,V \in \mathbb{R}^{|V|\times r}$ with $r \ll |V|$. This avoids explicitly materializing a dense $|V| \times |V|$ matrix and reduces the complexity to $O(|E|d + 2|V|rd)$.
>
> In addition to the complexity analysis, the appendix A.5.3 reports actual runtime comparisons, showing that the sparse inverse-operator variants reduce runtime by at least 45.0% compared with the dense explicit inverse operator.
>
> ## 5. Robustness to imperfect boundary definitions
>
> We agree that boundary specification is important for practical use. In the revised manuscript, we clarify that in our current setting boundary nodes are not manually annotated labels. They are determined by graph topology or by physical/geometric open boundaries. For directed river networks, upstream boundary nodes are zero-in-degree source nodes. For mesh-based or bidirectional physical graphs, boundary nodes are specified by physical or geometric open-boundary conditions.
>
> Nevertheless, we agree that imperfect boundary specification is an important practical concern. We now state this assumption explicitly and discuss that applications with uncertain domain boundaries would require additional robustness checks. Our experiments are conducted on observational datasets, which already contain measurement noise and environmental variability, but they do not exhaustively test adversarially missing, noisy, or incorrectly specified boundary nodes. We add this as a limitation and an important direction for future evaluation.